# Last-Iterate Convergence of Optimistic Gradient Method for Monotone Variational Inequalities

**Eduard Gorbunov**[*]
MIPT, Russia
Mila & UdeM, Canada
MBZUAI, UAE
eduard.gorbunov@mbzuai.ac.ae

**Adrien Taylor**
INRIA & École Normale Supérieure,
CNRS & PSL Research University, France
adrien.taylor@inria.fr

**Gauthier Gidel**
Mila & UdeM, Canada
Canada CIFAR AI Chair
gauthier.gidel@umontreal.ca

## Abstract

The Past Extragradient (PEG) [Popov, 1980] method, also known as the Optimistic Gradient method, has known a recent gain in interest in the optimization community with the emergence of variational inequality formulations for machine learning. Recently, in the unconstrained case, Golowich et al. [2020a] proved that a $\mathcal{O}(1/N)$ last-iterate convergence rate in terms of the squared norm of the operator can be achieved for Lipschitz and monotone operators with a Lipschitz Jacobian. In this work, by introducing a novel analysis through potential functions, we show that (i) this $\mathcal{O}(1/N)$ last-iterate convergence can be achieved without any assumption on the Jacobian of the operator, and (ii) it can be extended to the constrained case, which was not derived before even under Lipschitzness of the Jacobian. The proof is significantly different from the one known from Golowich et al. [2020a], and its discovery was computer-aided. Those results close the open question of the last iterate convergence of PEG for monotone variational inequalities.

## 1 Introduction

Minimax optimization, and more generally variational inequality problems, has known a surge of interest with the recent introduction of machine learning formulation with multiple objectives such as robust optimization [Ben-Tal et al., 2009], control [Hast et al., 2013], generative adversarial networks (GANs) [Goodfellow et al., 2014], and adversarial training [Goodfellow et al., 2015, Madry et al., 2018]. In this work, given a convex set $\mathcal{X}$, we focus on solving monotone variational inequalities:

$$\text{find } x^* \in \mathcal{X} \quad \text{such that} \quad \langle F(x^*), x - x^* \rangle \geq 0\,, \quad \forall x \in \mathcal{X} \subseteq \mathbb{R}^d. \quad \text{(VIP-C)}$$

In the unconstrained case ($\mathcal{X} = \mathbb{R}^d$,) this optimality condition can be simplified as

$$\text{find } x^* \in \mathbb{R}^d \quad \text{such that} \quad F(x^*) = 0. \quad \text{(VIP-U)}$$

We focus on the monotone and Lipschitz setting which is sufficient to ensure the existence of solutions for VIP-C and is the standard setting to study first-order methods [Facchinei and Pang, 2003].

**Assumption 1.** *$F : \mathcal{X} \to \mathbb{R}^d$ is monotone and $L$-Lipschitz, i.e., for all $x, y \in \mathcal{X}$*

$$\langle F(x) - F(y), x - y \rangle \geq 0 \qquad \text{and} \qquad \|F(x) - F(y)\| \leq L\|x - y\|, \quad (1)$$

*furthermore, there exists some $x^* \in \mathcal{X}$ which is a solution to (VIP-C).*

---

[*]The work was done when E. Gorbunov was researcher at MIPT and Mila & UdeM.

36th Conference on Neural Information Processing Systems (NeurIPS 2022).

In this context, it is well known that the standard Gradient method $x^{k+1} = \text{proj}[x^k - \gamma F(x^k)]$ (also known as the Forward method), where $\text{proj}(x) := \text{argmin}_{y \in \mathcal{X}} \|y - x\|^2$, does not always converge. Two important first-order methods have been introduced to circumvent this issue: the extragradient method (EG) [Korpelevich, 1976]

$$\widetilde{x}^k = \text{proj}[x^k - \gamma F(x^k)], \quad x^{k+1} = \text{proj}[x^k - \gamma F(\widetilde{x}^k)], \quad \text{for all } k > 0, \tag{Proj-EG}$$

and the past extragradient method (PEG) [Popov, 1980] defined via the following recursions: $\widetilde{x}^0 = x^0 \in \mathcal{X}$, $x^1 = \text{proj}[x^0 - \gamma F(x^0)]$, and

$$\widetilde{x}^k = \text{proj}[x^k - \gamma F(\widetilde{x}^{k-1})], \quad x^{k+1} = \text{proj}[x^k - \gamma F(\widetilde{x}^k)], \quad \text{for all } k > 0. \tag{Proj-PEG}$$

In the unconstrained case, the method simplifies to $x^1 = x^0 - \gamma F(x^0)$, and for all $k \geq 0$

$$\widetilde{x}^k = x^k - \gamma F(\widetilde{x}^{k-1}), \quad x^{k+1} = x^k - \gamma F(\widetilde{x}^k). \tag{PEG}$$

with the convention that $F(\widetilde{x}^{-1}) = 0$, allowing us to use the above recursions for $k = 0$. Note that, in this case, PEG is often studied in the equivalent form called Optimistic Gradient Method (OG):[2]

$$\widetilde{x}^{k+1} = \widetilde{x}^k - 2\gamma F(\widetilde{x}^k) + \gamma F(\widetilde{x}^{k-1}). \tag{OG}$$

Under Assumption 1, some recent works have managed to leverage tools for computer-aided proofs to show a $\mathcal{O}(1/N)$ (for the squared norm of the operator/squared residual[3]) last-iterate convergence rate for the extragradient method (EG), where $N$ denotes the total number of iterations. This was achieved in the unconstrained case by [Gorbunov et al., 2021] via the performance estimation technique [Drori and Teboulle, 2014, Taylor et al., 2017c] and in the constrained case [Cai et al., 2022] via Sum-of-Squares (SOS) techniques [Shor, 1987, Nesterov, 2000, Parrilo, 2000, Lasserre, 2001] (note that "performance estimation" corresponds to SOS of order 2). Our work leverages performance estimation problems (PEPs) for obtaining similar convergence results for PEG and Proj-PEG.

**Outline.** This paper is composed of five sections. In §1 we introduce our motivations, our main results and discuss the related work. In §2, we show how we used PEP to hint us toward a valid potential function. We prove the convergence of PEG in the *unconstrained case* and the *constrained cases* respectively in §3 and §4. Finally, we discuss our results and future research directions in §5. Since the proof in the unconstrained case is slightly simpler than in the constrained one (although it cannot be straightforwardly deduced from our analysis in the constrained case), in the main part of the paper, we discuss in detail the way to the proof in the unconstrained case, and defer the proofs and other details on the constrained case to Appendices C and D. Codes for verifying the potentials and convergence rates are publicly available: https://github.com/eduardgorbunov/potentials_and_last_iter_convergence_for_VIPs, the codes rely on the PEP packages [Taylor et al., 2017b, Goujaud et al., 2022] as well as on YALMIP [Lofberg, 2004].

## 1.1 Presentation of Our Main Results

For variational inequalities with possibly *unbounded* domains, there exist two main standard convergence criteria in the literature. The first one is the restricted gap function [Nesterov, 2007]

$$\text{Gap}_{F,R}(x^k) = \max_{y \in \mathcal{X}: \|y - x^*\| \leq R} \langle F(y), x^k - y \rangle, \tag{2}$$

where $x^*$ is any solution[4] of VIP-C. This quantity is an actual gap function only if $\|x^k - x^*\| \leq R$ and that it cannot be extended to the non-monotone setting where, for instance, there might exist several points where $F(x) = 0$. Since we show that $\|x^k - x^*\| \leq \frac{\sqrt{41}}{3}\|x^0 - x^*\|$ and $\|x^k - x^*\| \leq \sqrt{2}\|x^0 - x^*\| + \frac{\sqrt{2}}{L}\|F(x^0)\|$, $\forall k \geq 0$ in the unconstrained and constrained cases respectively, we can set $R = \frac{\sqrt{41}}{3}\|x^0 - x^*\|$ in the unconstrained case and $R = \sqrt{3}\|x^0 - x^*\| + \frac{1}{\sqrt{30}L}\|F(x^0)\|$, in

---

[2]See [Hsieh et al., 2019] for an overview of the different single-call variants of the extragradient method.

[3]By default, we always refer to the rates of convergence for $\|F(x^N)\|^2$ in the unconstrained case and $\|x^N - x^{N-1}\|^2$ in the constrained case.

[4]For simplicity, we slightly deviate from the standard notion of the restricted gap function from Nesterov [2007], since we do not assume that set $\{y \in \mathcal{X} \mid \|y - x^*\| \leq R\}$ contains all solutions of VIP-C. Taking $R$ larger than the diameter of the solution set resolves the discrepancy between definitions.

the constrained case and have that $\mathtt{Gap}_F(x^k) := \mathtt{Gap}_{F,R}(x^k)$ is a gap function for all $k \geq 0$. Another convergence criterion is the (squared) norm of the residual $\|x^{k+1} - x^k\|^2$. In the unconstrained setting, it is proportional to the (squared) norm of the operator $\|F(\widetilde{x}^k)\|^2$. This criterion is also valid as a local convergence certificate in the non-monotone case. We believe this criterion depicts a more precise picture than the standard gap function since it does not require to use a bound on $\|x^k - x^*\|$ to be valid and it generalizes to non-monotone settings. Nevertheless, we provide convergence results in terms of both criteria.

Our main theorems introduce new potential/Lyapunov functions for PEG with two main consequences: (i) it implies a uniform bound on $\|x^N - x^*\|$, and (ii) we show a $\mathcal{O}(1/\sqrt{N})$ convergence rate for $\|F(x^N)\|$ and $\mathtt{Gap}_F(x^N)$ in the unconstrained case and for $\|x^N - x^{N-1}\|$ and $\mathtt{Gap}_F(x^N)$ in the constrained case[5]. Theorem 1 and Theorem 2 below provide simplified versions of the results; more detailed/general statements are presented later in §3 and §4.

**Theorem 1** (Unconstrained Case). *Under Assumption 1, for all $N \geq 0$ and $\gamma = 1/3L$, we have*

$$\|F(x^N)\|^2 \leq \frac{123L^2\|x^0 - x^*\|^2}{N + 32}, \quad \mathtt{Gap}_F(x^N) \leq \frac{125L\|x^0 - x^*\|^2}{\sqrt{3N + 96}}, \tag{3}$$

*where $x^*$ is any solution to VIP-U.*

**Theorem 2** (Constrained Case). *Under Assumption 1, for all $N \geq 2$ the iterates of Proj-PEG with $\gamma = 1/4L$ satisfy*

$$\|x^N - x^{N-1}\|^2 \leq \frac{24H_0^2}{3N + 32}, \quad \mathtt{Gap}_F(x^N) \leq \frac{32L\sqrt{3}H_0^2}{\sqrt{3N + 32}}, \tag{4}$$

*where $H_0 > 0$ is such that $H_0^2 = 3\|x^0 - x^*\|^2 + \frac{1}{30L^2}\|F(x^0)\|^2$ and $x^*$ is any solution to VIP-C.*

## 1.2 Related Work

**Linear last-iterate convergence rates.** Motivated by nonconvex-nonconcave minimax formulation such as GANs, last-iterate convergences in the context of variational inequalities is the focus of many recent works. Linear convergence rates have been obtained, in the bilinear setting, the (local) strongly monotone setting, and similar settings such as sufficient bilinearity [Tseng, 1995, Daskalakis et al., 2018, Liang and Stokes, 2019, Gidel et al., 2019a,b, Mokhtari et al., 2019, Peng et al., 2020, Zhang and Yu, 2020, Abernethy et al., 2019, Loizou et al., 2020, Hsieh et al., 2019, Azizian et al., 2020a,b].

**Sublinear last-iterate convergence rates for EG and PEG.** More recently, the community has been focusing on the question of last-iterate convergence rate in the *monotone* setting (i.e., without strong monotonicity or sufficient bilinearity). For monotone and Lipschtiz operators obtaining $\mathcal{O}(1/N)$ last-iterate convergence rate of PEG was explicitly stated as an open question in [Hsieh et al., 2019]. In the unconstrained case, Golowich et al. [2020a] achieve this result by adding an assumption on the Jacobian of $F$ being Lipschitz. For EG as similar result has been obtained in Golowich et al. [2020b]. Eventually, a last-iterate convergence rate for EG has been provided by Gorbunov et al. [2021] in the unconstrained case and by Cai et al. [2022][6] in the constrained case, both under Assumption 1 solely. The question of last-iterate convergence rate for PEG both in the unconstrained and constrained cases mentioned by Hsieh et al. [2019] remained open until now and is the central question addressed here.

**Variants of EG and PEG.** Recently, some modification of EG with anchoring (a.k.a., Halpern iteration [Halpern, 1967]) enjoying (accelerated) last-iterate convergence rates have been proposed by [Yoon and Ryu, 2021], [Lee and Kim, 2021], and [Diakonikolas, 2020]. This work is concerned with method achieving the suboptimal $\mathcal{O}(1/N)$ rate, whereas $\mathcal{O}(1/N^2)$ can be achieved using optimal

---

[5]We notice that $\|x^N - x^{N-1}\| = \gamma\|F(\widetilde{x}^{N-1})\|$ in the unconstrained case, which differs from the quantity $\gamma\|F(x^N)\|$ that we estimate. However, $\|F(\widetilde{x}^k)\|$ and $\|F(x^k)\|$ are of comparable size since $\|F(\widetilde{x}^k) - F(x^k)\| \leq \gamma L\|F(\widetilde{x}^{k-1})\|$ for all $k \geq 0$.

[6]The paper [Cai et al., 2022] originally contained a $\mathcal{O}(1/N)$ analysis of the Extragradient method for Lipschitz monotone variational inequalities. In the updated version of their work, Cai et al. [2022] obtained $\mathcal{O}(1/N)$ convergence rates for OG using higher-order sum-of-squares. The results in our work were obtained independently and using a different approach. On the way, this work showcases that it is not necessary to use higher-order sum-of-squares when working with standard residual (i.e., using quadratic inequalities suffices).

methods [Diakonikolas, 2020, Yoon and Ryu, 2021, Lee and Kim, 2021, Tran-Dinh and Luo, 2021, Tran-Dinh, 2022]. We argue that (i) PEG is still largely used in practice, (ii) PEG is simple and more flexible, and (iii) that it benefits from additional advantageous properties, such as adaptivity to additional problem structure. Regarding (i) we would like to mention that Daskalakis et al. [2018] show that PEG-based algorithms (like PEG-Adam) perform well in training WGAN on CIFAR10 and PEG/OG have been extensively used in regret matching [Brown and Sandholm, 2019], counterfactual regret minimization [Farina et al., 2019], and for training agents to play poker [Anagnostides et al., 2022]. With a bit more details about (ii) and (iii): PEG is highly flexible and can be applied to online learning [Golowich et al., 2020a] or to the non-monotone variational inequalities [Daskalakis et al., 2018]. In contrast, EG with anchoring (EAG) [Yoon and Ryu, 2021]

$$\widetilde{x}^k = x^k + \beta_k \left( x^0 - x^k \right) - \gamma F(x^k), \quad x^{k+1} = x^k + \beta_k \left( x^0 - x^k \right) - \gamma F(x^k), \tag{EAG}$$

where $\beta_k \in [0, 1)$ is the anchoring coefficient, cannot be applied to online learning easily (EG/EAG are not no-regret [Golowich et al., 2020a]) and may not be desirable in the non-monotone setting since it may make some "bad" stationary points attractive.[7]

**Example 1.1.** *Let us consider a single example classification task with a deep linear neural network* $\min_{w \in \mathbb{R}^3} (y - w_3 w_2 w_1 x)^2 := f(w)$. *It has a undesirable stationary point* $w^s = (0, 0, 0)$. *Because* $\nabla^2 f(w^s) = 0$, *for an initialization* $w^0$ *close enough to* $w^s$ *and any small enough stepsize,* EAG *will converge to* $w^s$ *while* PEG *and* EG *will not, except for a zero measure set of initializations.*

A further practical reason that renders "simple" methods (such as PEG and EG) attractive is that simple methods are typically adaptive to additional problem structure (better behavior than predicted by the analysis when the problem has beneficial additional properties). As an example, PEG converges sublinearly for monotone operator (this is the topic of this work) and linearly for strongly-monotone operators [Gidel et al., 2019a] under the same stepsize rules. This stands in sharp contrast with optimal methods, which require to be tuned to the specific setting at hand (and in particular, which require the knowledge of the setting at hand).

## 2 A Path to the Proof

In this section, we show a direct approach for assessing the worst-case convergence rate of the last iterate of PEG. The Lyapunov analyses provided in the next sections are grounded on the numerical results presented here. Whereas converting those numerical results into a Lyapunov analysis is not direct, we believe that the material offers the comfortable privilege of a first clear $\mathcal{O}(1/N)$ baseline for the further convergence results, as well as a convenient approach for verifying Lyapunov functions.

**Verifying $\mathcal{O}(1/N)$ last-iterate convergence rate.** This section contains our heuristic argument for concluding a $\mathcal{O}(1/N)$ convergence of $\|F(x^N)\|^2$ for all $F$ satisfying our assumptions. This ingredient was the main motivation behind the investigations of the next sections. In short, our goal is to characterize the worst-case behavior of $\frac{\|F(x^N)\|^2}{\|x^0 - x^*\|^2}$ as a function of $N$ when $x^N$ is obtained from PEG. For doing that, we rely on the so-called performance estimation framework—first introduced in [Drori and Teboulle, 2014]. That is, we consider the problem of computing the worst-case value of $\frac{\|F(x^N)\|^2}{\|x^0 - x^*\|^2}$ (i.e., the worst possible $L$-Lipschitz and monotone $F$, worst dimension $d \in \mathbb{N}$, worst sequence of iterates $x^0, \ldots, x^N, \tilde{x}^0, \ldots, \tilde{x}^{N-1} \in \mathbb{R}^d$ and worst solution $x^* \in \mathbb{R}^d$ to (VIP-U)):

$$G_{\mathsf{PEG}}(\gamma, L, N) = \max_{\substack{F, d, x^* \\ \tilde{x}^0, \ldots, \tilde{x}^N \\ x^0, \ldots, x^N}} \frac{\|F(x^N)\|^2}{\|x^0 - x^*\|^2} \tag{5}$$

$$\begin{aligned} \text{s.t.} \quad & F \text{ is monotone and } L\text{-Lipschitz,} \\ & \tilde{x}^0 = x^0 \in \mathbb{R}^d, x^1 = x^0 - \gamma F(x^0) \\ & \widetilde{x}^k = x^k - \gamma F(\widetilde{x}^{k-1}), \text{ for } k = 1, \ldots, N, \\ & x^{k+1} = x^k - \gamma F(\widetilde{x}^k), \text{ for } k = 1, \ldots, N-1. \end{aligned}$$

---

[7]For example, stationary points $x^*$ such that $\text{Re}(\lambda) \gtrsim -\beta$, $\forall \lambda \in \text{Sp}(\nabla F(x^*))$ become locally attractive as long as the anchoring coefficient $\beta_k$ verifies $\beta_k \geq \beta$, which may correspond to an arbitrary long amount of time (see Example 1.1.)

Such problems are often referred to as Performance Estimation Problems (PEPs). Whereas it is not clear how to solve this PEP (5), a few techniques from [Ryu et al., 2020, Taylor et al., 2017a] allow obtaining a semidefinite relaxation providing meaningful worst-case bounds. By solving those problems numerically, we are able to *conjecture* that $G_{\mathsf{PEG}}(\gamma, L, N) = \mathcal{O}(1/N)$, as the convex relaxations provides upper bounds on $G_{\mathsf{PEG}}$ which appear to behave in $\mathcal{O}(1/N)$ in numerical experiments. More precisely, the convex relaxation under consideration arises from a sampled version of the previous problem (thereby passing from an infinite-dimensional problem to a finite-dimensional one). In other words, we consider a sampled version of $F$ with $g^k \approx F(x^k)$ and $\widetilde{g}^k \approx F(\widetilde{x}^k)$ (so the variables will be the iterates and the operators values at the iterates instead of the operator itself) and we require monotonicity and Lipschitzness to be satisfied at those points. For convenience, we define the set of samples $S = \{(x^*, 0)\} \cup \{(x^k, g^k)\}_{k=0}^N \cup \{(\widetilde{x}^k, \widetilde{g}^k)\}_{k=0}^N \subseteq \mathbb{R}^d \times \mathbb{R}^d$:

$$\widetilde{G}_{\mathsf{PEG}}(\gamma, L, N) = \max_{\substack{d \in \mathbb{N}, x^* \in \mathbb{R}^d \\ \{(x^k, g^k)\}_{k=0}^N \subset \mathbb{R}^d \times \mathbb{R}^d \\ \{(\widetilde{x}^k, \widetilde{g}^k)\}_{k=0}^N \subseteq \mathbb{R}^d \times \mathbb{R}^d}} \|g^N\|^2 \tag{6}$$

$$\text{s.t.} \quad \langle g - h, x - y \rangle \geq 0 \quad \forall (x, g), (y, h) \in S \tag{7}$$

$$\|g - h\|^2 \leq L^2 \|x - y\|^2 \quad \forall (x, g), (y, h) \in S \tag{8}$$

$$\widetilde{x}^0 = x^0 \in \mathbb{R}^d, x^1 = x^0 - \gamma g^0$$

$$\widetilde{x}^k = x^k - \gamma \widetilde{g}^{k-1}, \text{ for } k = 1, \ldots, N,$$

$$x^{k+1} = x^k - \gamma \widetilde{g}^k, \text{ for } k = 1, \ldots, N-1,$$

$$\|x^0 - x^*\|^2 \leq 1. \tag{9}$$

(Note that a classical homogeneity argument allows replacing the objective of (5) by $\|g^N\|^2$ plus the constraint $\|x^0 - x^*\|^2 \leq 1$; see, e.g., [Ryu et al., 2020, §3.1.1.].) In other words, we replace the constraint corresponding to the existence of monotone $L$-Lipschitz operator $F$ from (5) by the constraint that there exist sequences of points $\{x^k\}_{k=0}^N, \{\widetilde{x}^k\}_{k=0}^N$ and $\{g^k\}_{k=0}^N, \{\widetilde{g}^k\}_{k=0}^N$ such that they satisfy *necessary* conditions for the existence of monotone and $L$-Lipschitz operator $F$ interpolating them. Unfortunately, these constraints are not sufficient to ensure that there exists such monotone $L$-Lipschitz operator $F$ [Ryu et al., 2020]. That is, we can guarantee only that $\widetilde{G}_{\mathsf{PEG}}(\gamma, L, N) \geq G_{\mathsf{PEG}}(\gamma, L, N)$. Finallly, $\widetilde{G}_{\mathsf{PEG}}(\gamma, L, N)$ can be computed numerically using semidefinite programming (SDP) through standard solvers [Mosek, 2010, Sturm, 1999]. For formulating the computation of $\widetilde{G}_{\mathsf{PEG}}(\gamma, L, N)$ as an SDP we first substitute a number of variables: $\{x^k\}_{k=1}^N$ and $\{\widetilde{x}^k\}_{k=0}^N$ are linear combinations of $x^0, \{g^k\}_{k=0}^N, \{\widetilde{g}^k\}_{k=1}^N$. Next, we notice that the objective and constraints of problem (6) are linear functions of all possible inner products of vectors from $\mathbf{V} \overset{\text{def}}{=} (x^*, x^0, g^0, \widetilde{g}^1, g^1, \widetilde{g}^2, g^2, \ldots, \widetilde{g}^N, g^N)$. That is, problem (6) is linear w.r.t. the elements of Gram matrix $\mathbf{G} = \mathbf{V}^\top \mathbf{V} \succeq 0$ (we use the notation $\mathbf{G} \in \mathbb{S}_+^{2N+3}$ for denoting $(2N + 3) \times (2N + 3)$ symmetric positive semidefinite matrices). The problem also naturally features the constraint $\text{rank}(\mathbf{G}) \leq 2N + 3$, which becomes void due to maximization over the dimension $d$ in (6) (see, e.g., [Taylor et al., 2017c, Theorem 5]) and the $\widetilde{G}_{\mathsf{PEG}}(\gamma, L, N)$ can therefore be computed by solving a standard SDP:

$$\widetilde{G}_{\mathsf{PEG}}(\gamma, L, N) = \max_{\mathbf{G} \in \mathbb{S}_+^{2N+3}} \text{Tr}(\mathbf{M}_0 \mathbf{G}) \tag{10}$$

$$\text{s.t.} \quad \text{Tr}(\mathbf{M}_i \mathbf{G}) \leq 0 \text{ for } i = 1, 2, \ldots, 2N(2N + 1) + 1,$$

$$\text{Tr}(\mathbf{M}_{-1} \mathbf{G}) \leq 1,$$

where $\{\mathbf{M}_i\}_{i=-1}^{2N(2N+1)-1}$ are symmetric matrices encoding the objective and constraints from (6)–(9). For compactness, we omit the exact formulas for these matrices and refer to the examples for different PEPs from, e.g., [Ryu et al., 2020, Gorbunov et al., 2021].

By solving (10) numerically, we empirically observe that $\widetilde{G}_{\mathsf{PEG}}(\gamma, L, N) = \mathcal{O}(1/N)$ for different choices of $\gamma$, see Fig. 1a. Taking into account the lower bound $G_{\mathsf{PEG}}(\gamma, L, N) = \Omega(1/N)$ from Golowich et al. [2020a] we conclude that it is very likely that $\|F(x^N)\|^2 \sim 1/N$ for the values of $\gamma$ and $N$ under consideration. Of course, this observation is not a rigorous mathematical proof for the $\mathcal{O}(1/N)$ last-iterate convergence of PEG for two main reasons: (i) the SDP solver only outputs approximate SDP certificates (though highly accurate), and (ii) even by using exact SDP solvers (see, e.g., Henrion et al. [2016]), the worst-case bounds would only be valid for the values of the

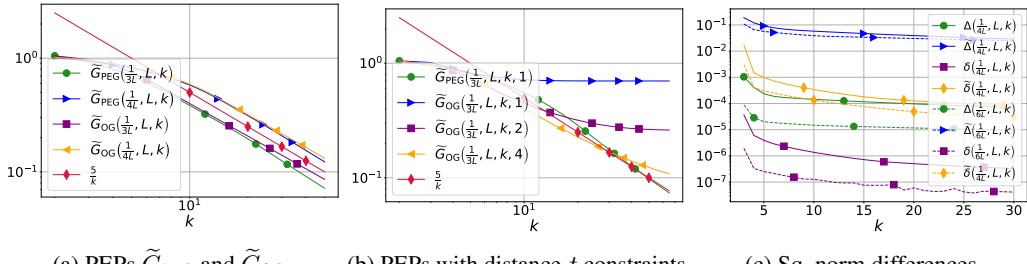

(a) PEPs $\widetilde{G}_{\mathsf{PEG}}$ and $\widetilde{G}_{\mathsf{OG}}$.     (b) PEPs with distance-$t$ constraints     (c) Sq. norm differences

Figure 1: In (a), we report $\widetilde{G}_{\mathsf{PEG}}(\gamma, L, N)$ and $\widetilde{G}_{\mathsf{OG}}(\gamma, L, N)$ for different values of $\gamma$ and $N$. In both cases, we observe $\mathcal{O}(1/N)$ convergence. In (b), we report $\widetilde{G}_{\mathsf{PEG}}(\gamma, L, N, 1)$ and $\widetilde{G}_{\mathsf{OG}}(\gamma, L, N, t)$ for $t = 1, 2, 4$. It suggest that $\widetilde{G}_{\mathsf{PEG}}(\gamma, L, N, 1) \sim 1/N$ but not $\widetilde{G}_{\mathsf{OG}}(\gamma, L, N, t)$ (even for $t = 4$). Finally, (c) shows how $\Delta(\gamma, L, N), \widetilde{\Delta}(\gamma, L, N), \delta(\gamma, L, N), \widetilde{\delta}(\gamma, L, N)$ evolve as $N$ grows.

parameters that were tried numerically; in particular, we only solved the SDPs for a few values of $N$. To translate those into rigorous worst-case bounds that are valid beyond the numerical trials, our goal is to identify feasible solution to the dual problem to (10) (each of those dual solutions corresponds to a valid upper bound on $\widetilde{G}_{\mathsf{PEG}}(\gamma, L, N)$, and thereby also on $G_{\mathsf{PEG}}(\gamma, L, N)$). One can find examples of such dual certificates in, e.g., [De Klerk et al., 2017, Taylor et al., 2017a].

**Why do we use PEG form instead of OG form?** It is relatively simple to verify that PEG and OG are equivalent [Gidel et al., 2019a, Hsieh et al., 2019]. Moreover, OG can be seen as a simplified version of PEG since OG explicitly uses only one sequence of points, while PEG relies on two sequences. As for PEG, one can consider a PEP for OG:

$$G_{\mathsf{OG}}(\gamma, L, N) = \max_{\substack{F, d, x^* \\ \widetilde{x}^0, \dots, \widetilde{x}^N}} \frac{\|F(\widetilde{x}^N)\|^2}{\|\widetilde{x}^0 - x^*\|^2} \tag{11}$$

$$\text{s.t.} \quad F \text{ is monotone and } L\text{-Lipschitz,}$$
$$\widetilde{x}^0 \in \mathbb{R}^d, \ \widetilde{x}^1 = \widetilde{x}^0 - \gamma F(\widetilde{x}^0),$$
$$\widetilde{x}^{k+1} = \widetilde{x}^k - 2\gamma F(\widetilde{x}^k) + \gamma F(\widetilde{x}^{k-1}), \text{ for } k = 1, \dots, N-1,$$

as well as its sampled version and the corresponding SDP relaxation, denoted by $\widetilde{G}_{\mathsf{OG}}(\gamma, L, N)$. Because its sampled version naturally contains only samples of $F$ at the iterates $\{\widetilde{x}^k\}_{k=0}^N$, the corresponding SDP has about 4 times less constraints than (10) (recall that (10) also involves Lipschitz and monotonicity inequalities with the iterates $\{x^k\}_{k=0}^N$). As for $\widetilde{G}_{\mathsf{PEG}}(\gamma, L, N)$, one can also compute $\widetilde{G}_{\mathsf{OG}}(\gamma, L, N)$ numerically using SDP solvers, see Fig. 1a. Those numerical results also suggest that $\widetilde{G}_{\mathsf{OG}}(\gamma, L, N) = \mathcal{O}(1/N)$ for the different choices of $\gamma$ provided in Fig. 1a, and $\widetilde{G}_{\mathsf{PEG}}(\gamma, L, N)$ and $\widetilde{G}_{\mathsf{OG}}(\gamma, L, N)$ are close to each other for the set of parameters under consideration. Finally, due to the fact that the SDP formulation of $\widetilde{G}_{\mathsf{OG}}(\gamma, L, N)$ involves less constraints than that of $\widetilde{G}_{\mathsf{PEG}}(\gamma, L, N)$ while providing very similar results, one might expect that studying $\widetilde{G}_{\mathsf{OG}}(\gamma, L, N)$ might be simpler.

Convergence proofs of classical first-order optimization methods generally rely on clever combinations of inequalities characterizing the class of problems at hand, and the algorithm. Those inequalities typically involve consecutive iterates and/or a solution $x^*$ to the variational problem. For finding a (hopefully) simple proof, let us introduce a few relaxations of $\widetilde{G}_{\mathsf{PEG}}(\gamma, L, N)$ and $\widetilde{G}_{\mathsf{OG}}(\gamma, L, N)$, parameterized by some $t \in \mathbb{N}$ and respecitvely denoted by $\widetilde{G}_{\mathsf{PEG}}(\gamma, L, N, t)$ and $\widetilde{G}_{\mathsf{OG}}(\gamma, L, N, t)$. Those values respectively correspond to the optimal values of the respective SDP formulations of $\widetilde{G}_{\mathsf{PEG}}$ and $\widetilde{G}_{\mathsf{OG}}$ where we removed all constraints corresponding to pairs of iterates $(i, j)$ with $|i - j| > t$. We refer to the remaining constraints as *distance-$t$ constraints*. For example, distance-1 constraints involve monotonicity and Lipschitzness inequalities between two consecutive iterates as well as between the iterates and the solution $x^*$ under consideration. As provided by Fig. 1b, solving the corresponding SDPs for PEG and OG numerically, we observe that $\widetilde{G}_{\mathsf{PEG}}(\gamma, L, N, 1) \sim 1/N$, but the value of $\widetilde{G}_{\mathsf{OG}}(\gamma, L, N, 1)$ seem to stall after a few iterations. This experiment suggests

necessity of using the values of $F$ evaluated at $\{x^k\}_{k \geq 0}$ for obtaining "simple" proofs involving only inequalities with consecutive iterates.

**The norm does not decrease monotonically.** Previous numerical experiments suggest that there exist a proof of $\mathcal{O}(1/N)$ last-iterate convergence of PEG which uses only inequalities involving consecutive iterates and $x^*$. However, the problem of finding the proof remains somehow involved and we did not manage to find analytical expressions (as functions of $\gamma$, $L$, and $N$) to the dual SDP formulation to $\widetilde{G}_{\mathsf{PEG}}(\gamma, L, N, 1)$. Therefore, the following lines aim at finding appropriate Lyapunov function, that is, even looser upper bounds on $G_{\mathsf{PEG}}(\gamma, L, N)$. As a starting point, it is shown in [Gorbunov et al., 2021] that the extragradient method (EG) satisfies $\|F(x^{k+1})\|^2 \leq \|F(x^k)\|^2$ for all $k \geq 0$ and all monotone Lipschitz operator $F$. Since EG and PEG are very similar methods, it is natural to check whether the same inequality holds for PEG. One way to verify whether this inequality holds is to check nonpositivity of the following PEP:

$$\max_{\substack{F, d, x^* \\ \tilde{x}^0, \ldots, \tilde{x}^N \\ x^0, \ldots, x^N}} \frac{\|F(x^{N+1})\|^2 - \|F(x^N)\|^2}{\|x^0 - x^*\|^2} \tag{12}$$

$$\begin{aligned}
\text{s.t.} \quad & F \text{ is monotone and } L\text{-Lipschitz}, \\
& \tilde{x}^0 = x^0 \in \mathbb{R}^d, \ x^1 = x^0 - \gamma F(x^0), \\
& \tilde{x}^k = x^k - \gamma F(\tilde{x}^{k-1}), \text{ for } k = 1, \ldots, N, \\
& x^{k+1} = x^k - \gamma F(\tilde{x}^k), \text{ for } k = 1, \ldots, N-1,
\end{aligned}$$

through its SDP relaxation, whose value is denoted by $\Delta(\gamma, L, N)$. A similar SDP can be formulated for the objective $\|F(\tilde{x}^N)\|^2 - \|F(\tilde{x}^{N-1})\|^2$ and we denote its corresponding optimal value by $\widetilde{\Delta}(\gamma, L, N)$. For convenience, we also define two corresponding SDP formulation whose optimal values are denoted by $\delta(\gamma, L, N)$ and $\widetilde{\delta}(\gamma, L, N)$ and which corresponds to respectively the SDP formulations of $\Delta(\gamma, L, N)$ and $\widetilde{\Delta}(\gamma, L, N)$ where the Lipschitz and monotonicity constraints are replaced by cocoercivity (i.e., $\langle g - h, x - y \rangle \geq 0$ and $\|g - h\|^2 \leq L^2 \|x - y\|^2$ are replaced by $\|g - h\|^2 \leq L\langle g - h, x - y \rangle$). Cocoercive operators are both Lipschitz and monotone, and one advantageous property of $\delta(\gamma, L, N)$ and $\widetilde{\delta}(\gamma, L, N)$ is that they are guaranteed to be "tight". That is, one can construct operators matching the numerical values of respectively $\frac{\|F(x^{N+1})\|^2 - \|F(x^N)\|^2}{\|x^0 - x^*\|^2}$ and $\frac{\|F(\tilde{x}^{N+1})\|^2 - \|F(\tilde{x}^N)\|^2}{\|\tilde{x}^0 - x^*\|^2}$ obtained from the SDPs, see [Ryu et al., 2020, Proposition 2]. Therefore a positive value for $\delta(\gamma, L, N)$ (resp. $\widetilde{\delta}(\gamma, L, N)$) implies the existence of some $L$-Lipschitz monotone $F$ satisfying $\|F(x^{N+1})\|^2 \geq \|F(x^N)\|^2$ (resp. $\|F(\tilde{x}^{N+1})\|^2 \geq \|F(\tilde{x}^N)\|^2$). Therefore, Fig. 1c allows concluding that $\|F(x^N)\|^2$ (resp. $\|F(\tilde{x}^N)\|^2$) is not a decreasing function of $N$ in general. This fact highlights the difference between EG and PEG in terms of the analysis. However, this experiment also suggests that $\Delta(\gamma, L, N)$ is a decreasing function of $N$ (for some values of $\gamma$).

Therefore, although $\|F(x^N)\|^2$ is not a decreasing function of $N$ for all monotone $L$-Lipschitz operator $F$ and all starting point $x_0 \in \mathbb{R}^d$, it appears that the difference $\|F(x^{N+1})\|^2 - \|F(x^N)\|^2$ decrease to zero (for all starting point and all operator $F$ satisfying our assumptions) as $N$ grows. Those numerical results suggest that there is a chance to find a non-negative sequence $\{A_N\}_{N \geq 0}$ for which $\|F(x^{N+1})\|^2 + A_{N+1} \leq \|F(x^N)\|^2 + A_N$. To discover such a sequence, we carefully studied the numerical values of the solutions to the dual SDP for different values of $\gamma$ and $N$ and tried to infer some structure from them. The following section reports the positive results in that direction.

## 3 Last Iterate Convergence of PEG in the Unconstrained Case

The technique described in the previous section allowed performing an educated trial and error procedure, searching for appropriate potential functions. We finally found a suitable candidate $A_N$. In particular, one can numerically check that $\|F(x^{N+1})\|^2 + 2\|F(x^{N+1}) - F(\tilde{x}^N)\|^2 \leq \|F(x^N)\|^2 + 2\|F(x^N) - F(\tilde{x}^{N-1})\|^2$ for all monotone $L$-Lipschitz $F$ for all tested values of $N$ and $\gamma \leq \sqrt{2}/3L$. Using the few nonzero dual variables for the problem of verifying this potential function allows reaching the desired proof after a bit of manual cleaning.

**Lemma 3.1.** *Under Assumption 1, the iterates of* PEG *with* $\gamma > 0$ *satisfy for any* $k > 0$,

$$\|F(x^{k+1})\|^2 + 2\|F(x^{k+1}) - F(\widetilde{x}^k)\|^2 \leq \|F(x^k)\|^2 + 2\|F(x^k) - F(\widetilde{x}^{k-1})\|^2$$
$$+ 3\left(L^2\gamma^2 - \frac{2}{9}\right)\|F(\widetilde{x}^k) - F(\widetilde{x}^{k-1})\|^2. \qquad (13)$$

*We notice that the last term is non-positive for* $0 < \gamma \leq \sqrt{2}/3L$.

*Proof.* Since $F$ is monotone and $L$-Lipschitz, we have

$$0 \leq \langle F(x^{k+1}) - F(x^k), x^{k+1} - x^k\rangle,$$
$$\|F(x^{k+1}) - F(\widetilde{x}^k)\|^2 \leq L^2\|x^{k+1} - \widetilde{x}^k\|^2.$$

By definition of $x^{k+1}$ and $\widetilde{x}^k$, we have $x^{k+1} - x^k = -\gamma F(\widetilde{x}^k)$ and $x^{k+1} - \widetilde{x}^k = x^k - \gamma F(\widetilde{x}^k) - x^k + \gamma F(\widetilde{x}^{k-1}) = \gamma\left(F(\widetilde{x}^{k-1}) - F(\widetilde{x}^k)\right)$. Plugging these relations to the above inequalities and dividing the first inequality by $\gamma$, we get

$$0 \leq \langle F(x^k) - F(x^{k+1}), F(\widetilde{x}^k)\rangle,$$
$$\|F(x^{k+1}) - F(\widetilde{x}^k)\|^2 \leq L^2\gamma^2\|F(\widetilde{x}^k) - F(\widetilde{x}^{k-1})\|^2.$$

Next, we sum up the above inequalities with weights $\lambda_1 = 2$ and $\lambda_2 = 3$ respectively:

$$\begin{aligned}
3\|F(x^{k+1}) - F(\widetilde{x}^k)\|^2 &\leq 2\langle F(x^k) - F(x^{k+1}), F(\widetilde{x}^k)\rangle + 3L^2\gamma^2\|F(\widetilde{x}^k) - F(\widetilde{x}^{k-1})\|^2 \\
&\overset{(20)}{=} \|F(x^k)\|^2 + \|F(\widetilde{x}^k)\|^2 - \|F(x^k) - F(\widetilde{x}^k)\|^2 \\
&\quad + \|F(x^{k+1}) - F(\widetilde{x}^k)\|^2 - \|F(x^{k+1})\|^2 - \|F(\widetilde{x}^k)\|^2 \\
&\quad + 3L^2\gamma^2\|F(\widetilde{x}^k) - F(\widetilde{x}^{k-1})\|^2.
\end{aligned}$$

Rearranging the terms, we derive

$$\begin{aligned}
\|F(x^{k+1})\|^2 + 2\|F(x^{k+1}) - F(\widetilde{x}^k)\|^2 &\leq \|F(x^k)\|^2 - \|F(x^k) - F(\widetilde{x}^k)\|^2 \\
&\quad + 3L^2\gamma^2\|F(\widetilde{x}^k) - F(\widetilde{x}^{k-1})\|^2. \qquad (14)
\end{aligned}$$

To estimate the negative term from the right-hand side of the above inequality we use that $-\|a - b\|^2 \overset{(22)}{\leq} -\frac{1}{1+\alpha}\|a\|^2 + \frac{1}{\alpha}\|b\|^2$ with $a = F(\widetilde{x}^k) - F(\widetilde{x}^{k-1})$, $b = F(x^k) - F(\widetilde{x}^{k-1})$ and $\alpha = 1/2$:

$$-\|F(x^k) - F(\widetilde{x}^k)\|^2 \leq -\frac{2}{3}\|F(\widetilde{x}^{k-1}) - F(\widetilde{x}^k)\|^2 + 2\|F(x^k) - F(\widetilde{x}^{k-1})\|^2.$$

Plugging the above inequality in (14), we obtain

$$\begin{aligned}
\|F(x^{k+1})\|^2 + 2\|F(x^{k+1}) - F(\widetilde{x}^k)\|^2 &\leq \|F(x^k)\|^2 + 2\|F(x^k) - F(\widetilde{x}^{k-1})\|^2 \\
&\quad + 3\left(L^2\gamma^2 - \frac{2}{9}\right)\|F(\widetilde{x}^k) - F(\widetilde{x}^{k-1})\|^2,
\end{aligned}$$

which finishes the proof. $\qquad\square$

Using this lemma we then show the last-iterate convergence rate of PEG.

**Theorem 1.** *Let operator* $F$ *be monotone and* $L$-Lipschitz. *Then for all* $k \geq 0$ *the iterates of* PEG *with* $\gamma \leq 1/3L$ *satisfy* $\Phi_{k+1} \leq \Phi_k$ *with* $\Phi_k$ *defined as*

$$\Phi_k = \|x^k - x^*\|^2 + \frac{k+32}{3}\gamma^2\left(\|F(x^k)\|^2 + 2\|F(x^k) - F(\widetilde{x}^{k-1})\|^2\right). \qquad (15)$$

*In particular, for all* $N \geq 0$ *and* $\gamma \leq 1/3L$ *the above formula implies*

$$\|F(x^N)\|^2 \leq \frac{3(1 + 32L^2\gamma^2)\|x^0 - x^*\|^2}{\gamma^2(N+32)}, \quad \textit{Gap}_F(x^k) \leq \frac{2\sqrt{41}(1 + 32L^2\gamma^2)\|x^0 - x^*\|^2}{\gamma\sqrt{3N+96}}. \qquad (16)$$

As we write before, in contrast to Golowich et al. [2020a], our results does not rely on the Lipschitzness of $\nabla F(x)$. Moreover, even when $\nabla F(x)$ is assumed to be $\Lambda$-Lipschitz, our result improve upon Golowich et al. [2020a] in certain regimes. In particular, if $\Lambda\|x^0 - x^*\| \gg L$, which is typically the case (e.g., see Appendix B from Gorbunov et al. [2021] for the details), the constants in our upper bounds are significantly smaller and even when $\Lambda\|x^0 - x^*\| \ll L$ our result allows for stepsizes 50 times larger with a improvement by a factor $\approx 10^6$ in terms of convergence rate for $\|F(x^k)\|^2$.

Next, our analysis significantly differs from the previous known one from Golowich et al. [2020a]. In particular, using Lipschitzness of Jacobian, Golowich et al. [2020a] derive that $\|F(\widetilde{x}^N)\|^2$ is not much larger than $\min_{k=0,\ldots,N}\|F(x^k)\|^2$, which instantly implies a $\mathcal{O}(1/N)$ last-iterate convergence rate after applying standard results for the best-iterate convergence of PEG. In contrast, we do not derive any connections between the best and the last iterates of PEG and directly rely on the fact that $\|F(x^k)\|^2 + 2\|F(x^k) - F(\widetilde{x}^{k-1})\|^2$ is a decreasing function of $k$, i.e., on Lemma 3.1.

We also remark that the derived upper bounds are worse than those that we obtained numerically and reported in Fig. 1a. For example, when $N = 50$ and $\gamma = 1/3L$ the upper bound on $\|F(x^N)\|^2$ from Theorem 1 is $\approx 20$ times worse than the upper bound found numerically. Since our goal was in deriving a *simple* proof of the $\mathcal{O}(1/N)$ rate, we did not try to improve the multiplicative constants in the final results. We also do not plot the curve corresponding to the upper bound on $\|F(x^N)\|^2$ from Theorem 1 for the sake of readability of Fig. 1a.

## 4    Last Iterate Convergence of PEG in the Constrained Case

In this section, we extend the last-iterate convergence rates proven in Theorem 1 to the constrained case. Such an extension is not straightforward because the convergence criterion considered in the unconstrained case cannot be used anymore in the constrained case. That is, Lemma 3.1 is not valid anymore and cannot be easily adapted to the constrained case: norm of the operator $\|F(x^k)\|$ does not necessarily converge to zero since $F(x^*) \neq 0$ in general. However, to find new potential function we used the same techniques as in the unconstrained case. This search led us to the following result.

**Lemma 4.1.** *Under Assumption 1, the iterates of* Proj-PEG *with* $\gamma > 0$ *satisfy for any* $k > 0$,

$$\Psi_{k+1} \leq \Psi_k - \left(1 - 5L^2\gamma^2\right)\|x^{k+1} - \widetilde{x}^k\|^2 - \gamma^2\|F(x^{k+1}) - F(\widetilde{x}^k)\|^2, \qquad (17)$$

*where* $\Psi_k = \|x^k - x^{k-1}\|^2 + \|x^k - x^{k-1} - 2\gamma(F(x^k) - F(\widetilde{x}^{k-1}))\|^2$.

*Proof sketch.* Numerically we discovered that to prove this result it is sufficient to sum up constraints corresponding to the monotonicity at $(x^k, x^{k+1})$, Lipschitzness at $(x^{k+1}, \widetilde{x}^k)$, and projections properties[8] for the pairs $(x^{k+1}, x^k), (x^k, x^{k+1}), (x^k, \widetilde{x}^k), (\widetilde{x}^k, x^{k+1})$, with weights $4\gamma, 5\gamma^2, 4, 2, 2, 2$ respectively. After that, it remains to rearrange the terms and apply standard inequalities from Appendix A to derive the result. See the detailed proof in Appendix C. $\qquad\square$

This lemma is critically different from Lemma 3.1 since in the unconstrained case the potential from Lemma 4.1 reduces to $\Psi_k = \|F(\widetilde{x}^{k-1})\|^2 + \|F(\widetilde{x}^{k-1}) - 2\gamma(F(x^k) - F(\widetilde{x}^{k-1}))\|^2$. Moreover, as one can see from the next result, using the potential from Lemma 4.1, we obtain the proof valid for slightly smaller stepsize than in the unconstrained case (see also Appendix D for the numerical verification of the rate).

**Theorem 2.** *Under Assumption 1, the iterates of* Proj-PEG *with* $\gamma \leq 1/4L$ *satisfy* $\Phi_{k+1} \leq \Phi_k$, $k \geq 2$ *with* $\Phi_k$ *defined as*

$$\Phi_k = \|x^k - x^*\|^2 + \frac{1}{16}\|\widetilde{x}^{k-1} - \widetilde{x}^{k-2}\|^2 + \frac{3k+32}{24}\Psi_k, \qquad (18)$$

*where* $\Psi_k = \|x^k - x^{k-1}\|^2 + \|x^k - x^{k-1} - 2\gamma(F(x^k) - F(\widetilde{x}^{k-1}))\|^2$. *In particular, it implies*

$$\|x^N - x^{N-1}\|^2 \leq \frac{24H_{0,\gamma}^2}{3N+32}, \quad \mathit{Gap}_F(x^N) \leq \frac{8\sqrt{3}H_{0,\gamma} \cdot H_0}{\gamma\sqrt{3N+32}}, \quad \forall N \geq 2, \qquad (19)$$

*where* $H_0, H_{0,\gamma} > 0$ *are such that* $H_{0,\gamma}^2 = 2(1 + 3\gamma^2L^2 + 4\gamma^4L^4)\|x^0 - x^*\|^2 + \left(\frac{41}{12} + \frac{19}{3}\gamma^2L^2\right)\gamma^2\|F(x^0)\|^2$, $H_0^2 = 3\|x^0 - x^*\|^2 + \frac{1}{30L^2}\|F(x^0)\|^2$.

---

[8]By projection property for the pair $(x_+, y)$, where $x_+ = \text{proj}[x]$, $y \in \mathcal{X}$, we mean $\langle x - x_+, y - x_+ \rangle \leq 0$.

This result extends last-iterate convergence of Proj-PEG to the constrained case. The closest result in the literature is Cai et al. [2022, Theorem 3] that gives a last-iterate convergence result for EG in terms of the tangent residual and gap function. Moreover, we show a $\mathcal{O}(1/N)$ last-iterate convergence result for the residuals, which is stronger than the result in Cai et al. [2022, Theorem 3]. Moreover, our results justify that higher-order SOS programs (with polynomials of degree larger than 2) used by Cai et al. [2022] are not required to derive tight last-iterate convergence results for Proj-PEG. We believe that this is the case for other EG-like methods and similar rates could be proven for the residuals of these methods as well (see Appendix D for preliminary results supporting this claim).

## 5  Discussion

In this work, we leveraged the performance estimation problem framework to show the last-iterate convergence of PEG of monotone and Lipschitz operators both in the constrained and unconstrained cases. These results answer some important questions that remained open until now in the variational inequality literature showcasing the appealing properties of extrapolation-based methods. However, important open questions remain such as last-iterate convergence rates for stochastic methods for monotone and Lipschitz variational inequalities or a better understanding of the dynamics in non-monotone cases that occur in machine learning applications. Finally, the results obtained for PEG in this work have worse multiplicative constants and more restrictive range of stepsizes than those obtained for EG Gorbunov et al. [2021], Cai et al. [2022]. It would be interesting to address this discrepancy in the future work.

## Acknowledgments and Disclosure of Funding

The authors would like to warmly thank Sylvain Sorin for spotting a typo in Appendix C, as well as for a few suggestions of improvements.

This work was partially supported by a grant for research centers in the field of artificial intelligence, provided by the Analytical Center for the Government of the Russian Federation in accordance with the subsidy agreement (agreement identifier 000000D730321P5Q0002) and the agreement with the Moscow Institute of Physics and Technology dated November 1, 2021 No. 70-2021-00138. A. Taylor acknowledges support from the French "Agence Nationale de la Recherche" as part of the "Investissements d'avenir" program, reference ANR-19-P3IA-0001 (PRAIRIE 3IA Institute), as well as support from the European Research Council (grant SEQUOIA 724063).

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
