# A  Useful Facts

**Standard inequalities.** For all $a, b \in \mathbb{R}^d$ and $\alpha > 0$ the following relations hold:

$$2\langle a, b \rangle \quad = \quad \|a\|^2 + \|b\|^2 - \|a - b\|^2, \tag{20}$$

$$\|a + b\|^2 \quad \leq \quad (1 + \alpha)\|a\|^2 + (1 + \alpha^{-1})\|b\|^2, \tag{21}$$

$$-\|a - b\|^2 \quad \leq \quad -\frac{1}{1 + \alpha}\|a\|^2 + \frac{1}{\alpha}\|b\|^2. \tag{22}$$

**Auxiliary results.** In the analysis of Proj-PEG, we use two lemmas from Gidel et al. [2019a].

**Lemma A.1** (Lemma 5 from Gidel et al. [2019a]). *Let operator $F$ be $L$-Lipschitz. Then for all $k \geq 1$ the iterates of* Proj-PEG *with $\gamma > 0$ satisfy*

$$2\gamma\langle F(\widetilde{x}^k), \widetilde{x}^k - x^* \rangle \leq \|x^k - x^*\|^2 - \|x^{k+1} - x^*\|^2 - \|\widetilde{x}^k - x^k\|^2 + \gamma^2 L^2\|\widetilde{x}^k - \widetilde{x}^{k-1}\|^2. \tag{23}$$

**Lemma A.2** (Lemma 6 from Gidel et al. [2019a]). *Let operator $F$ be $L$-Lipschitz. Then for all $k \geq 2$ the iterates of* Proj-PEG *with $\gamma > 0$ satisfy*

$$\|\widetilde{x}^k - \widetilde{x}^{k-1}\|^2 \leq 4\|\widetilde{x}^k - x^k\|^2 + 4\gamma^2 L^2\|\widetilde{x}^{k-1} - \widetilde{x}^{k-2}\|^2 - \|\widetilde{x}^k - \widetilde{x}^{k-1}\|^2. \tag{24}$$

# B  Proof of Theorem 1

Let us first recall Theorem 1.

**Theorem 1.** *Let operator $F$ be monotone and $L$-Lipschitz. Then for all $k \geq 0$ the iterates of PEG with $\gamma \leq 1/3L$ satisfy $\Phi_{k+1} \leq \Phi_k$ with $\Phi_k$ defined as*

$$\Phi_k = \|x^k - x^*\|^2 + \frac{k+32}{3}\gamma^2\left(\|F(x^k)\|^2 + 2\|F(x^k) - F(\widetilde{x}^{k-1})\|^2\right). \tag{25}$$

*In particular, for all $N \geq 0$ and $\gamma \leq 1/3L$ the above formula implies*

$$\|F(x^N)\|^2 \leq \frac{3(1 + 32L^2\gamma^2)\|x^0 - x^*\|^2}{\gamma^2(N+32)}, \quad \mathit{Gap}_F(x^k) \leq \frac{2\sqrt{41}(1+32L^2\gamma^2)\|x^0 - x^*\|^2}{\gamma\sqrt{3N+96}}. \tag{26}$$

*Proof.* We start with the upper bound for $\|x^{k+1} - x^*\|^2$:

$$
\begin{aligned}
\|x^{k+1} - x^*\|^2 &= \|x^k - x^* - \gamma F(\widetilde{x}^k)\|^2 \\
&= \|x^k - x^*\|^2 - 2\gamma\langle x^k - x^*, F(\widetilde{x}^k)\rangle + \gamma^2\|F(\widetilde{x}^k)\|^2 \\
&= \|x^k - x^*\|^2 - 2\gamma\langle \widetilde{x}^k - x^*, F(\widetilde{x}^k)\rangle - 2\gamma\langle x^k - \widetilde{x}^k, F(\widetilde{x}^k)\rangle + \gamma^2\|F(\widetilde{x}^k)\|^2.
\end{aligned}
$$

Since $F$ is monotone, we have $\langle \widetilde{x}^k - x^*, F(\widetilde{x}^k)\rangle \geq 0$. Moreover, the update rule for PEG implies $\langle x^k - \widetilde{x}^k, F(\widetilde{x}^k)\rangle = \gamma\langle F(\widetilde{x}^{k-1}), F(\widetilde{x}^k)\rangle$. Using these relations, we continue our derivation as

$$
\begin{aligned}
\|x^{k+1} - x^*\|^2 &\leq \|x^k - x^*\|^2 - 2\gamma^2\langle F(\widetilde{x}^{k-1}), F(\widetilde{x}^k)\rangle + \gamma^2\|F(\widetilde{x}^k)\|^2 \\
&= \|x^k - x^*\|^2 + \gamma^2\|F(\widetilde{x}^{k-1}) - F(\widetilde{x}^k)\|^2 - \gamma^2\|F(\widetilde{x}^{k-1})\|^2.
\end{aligned}
$$

Next, we sum up the above inequality with $((k+33)\gamma^2/3)$-multiple of (13) and use the definition of $\Phi_k$ from (25):

$$
\begin{aligned}
\Phi_{k+1} \leq\ & \|x^k - x^*\|^2 + \gamma^2\|F(\widetilde{x}^{k-1}) - F(\widetilde{x}^k)\|^2 - \gamma^2\|F(\widetilde{x}^{k-1})\|^2 \\
& + \frac{(k+33)\gamma^2}{3}\left(\|F(x^k)\|^2 + 2\|F(x^k) - F(\widetilde{x}^{k-1})\|^2\right) \\
& + (k+33)\gamma^2\left(L^2\gamma^2 - \frac{2}{9}\right)\|F(\widetilde{x}^k) - F(\widetilde{x}^{k-1})\|^2 \\
\overset{\gamma \leq 1/3L}{\leq}\ & \Phi_k + \frac{\gamma^2}{3}\left(\|F(x^k)\|^2 + 2\|F(x^k) - F(\widetilde{x}^{k-1})\|^2\right) - \gamma^2\|F(\widetilde{x}^{k-1})\|^2 \\
& + \gamma^2\left(1 - \frac{k+33}{9}\right)\|F(\widetilde{x}^k) - F(\widetilde{x}^{k-1})\|^2.
\end{aligned}
$$

Applying $\|F(x^k)\|^2 \leq 2\|F(\widetilde{x}^{k-1})\|^2 + 2\|F(x^k) - F(\widetilde{x}^{k-1})\|^2$ and then $\|F(x^k) - F(\widetilde{x}^{k-1})\|^2 \leq 2\|F(x^k) - F(\widetilde{x}^k)\|^2 + 2\|F(\widetilde{x}^k) - F(\widetilde{x}^{k-1})\|^2 \overset{(1)}{\leq} 2L^2\|x^k - \widetilde{x}^k\|^2 + 2\|F(\widetilde{x}^k) - F(\widetilde{x}^{k-1})\|^2 = 2L^2\gamma^2\|F(\widetilde{x}^{k-1})\|^2 + 2\|F(\widetilde{x}^k) - F(\widetilde{x}^{k-1})\|^2 \leq \frac{2}{9}\|F(\widetilde{x}^{k-1})\|^2 + 2\|F(\widetilde{x}^k) - F(\widetilde{x}^{k-1})\|^2$, we derive

$$
\begin{aligned}
\Phi_{k+1} \leq\ & \Phi_k + \frac{8}{3}\gamma^2\|F(x^k) - F(\widetilde{x}^{k-1})\|^2 - \frac{\gamma^2}{3}\|F(\widetilde{x}^{k-1})\|^2 \\
& + \gamma^2\left(1 - \frac{k+33}{9}\right)\|F(\widetilde{x}^k) - F(\widetilde{x}^{k-1})\|^2 \\
\leq\ & \Phi_k + \gamma^2\left(\frac{8}{27} - \frac{1}{3}\right)\|F(\widetilde{x}^{k-1})\|^2 + \gamma^2\left(\frac{11}{3} - \frac{k+33}{9}\right)\|F(\widetilde{x}^k) - F(\widetilde{x}^{k-1})\|^2 \\
\leq\ & \Phi_k,
\end{aligned}
$$

where in the last step we use $k \geq 0$. In other words, we proved that $\Phi_k$ defined in (25) is a potential for (PEG). In particular, $\Phi_k \leq \Phi_{k-1} \leq \ldots \leq \Phi_0 = \|x^0 - x^*\|^2 + 32\gamma^2\|F(x^0)\|^2 \overset{(1)}{\leq} (1 + 32L^2\gamma^2)\|x^0 - x^*\|^2$ (here we use our convention: $F(\widetilde{x}^{-1}) = 0$) and all terms in $\Phi_k$ are non-negative. These facts imply that for all $k \geq 0$

$$
\begin{aligned}
\|F(x^k)\|^2 &\leq \frac{3}{\gamma^2(k+32)}\Phi_k \leq \frac{3(1+32L^2\gamma^2)\|x^0 - x^*\|^2}{\gamma^2(k+32)}, \tag{27} \\
\|x^k - x^*\|^2 &\leq \Phi_k \leq (1 + 32L^2\gamma^2)\|x^0 - x^*\|^2. \tag{28}
\end{aligned}
$$

That is, we obtained the first part of (26). The second part of (26) follows from the monotonicity of $F$ and the above inequalities:

$$
\begin{aligned}
\text{Gap}_F(x^k) \quad &= \quad \max_{y \in \mathbb{R}^d : \|y - x^*\| \leq \frac{\sqrt{41}}{3} \|x^0 - x^*\|} \langle F(y), x^k - y \rangle \\
&\overset{(1)}{\leq} \quad \max_{y \in \mathbb{R}^d : \|y - x^*\| \leq \frac{\sqrt{41}}{3} \|x^0 - x^*\|} \langle F(x^k), x^k - y \rangle \\
&\leq \quad \|F(x^k)\| \cdot \max_{y \in \mathbb{R}^d : \|y - x^*\| \leq \frac{\sqrt{41}}{3} \|x^0 - x^*\|} \|x^k - y\| \\
&\leq \quad \|F(x^k)\| \left( \|x^k - x^*\| + \frac{\sqrt{41}}{3} \|x^0 - x^*\| \right) \\
&\overset{(27),(28)}{\leq} \quad \frac{2\sqrt{41}(1 + 32L^2\gamma^2)\|x^0 - x^*\|^2}{\gamma\sqrt{3k + 96}}.
\end{aligned}
$$

$\square$

## C  Proof of Theorem 2

Let us start with an important lemma.

**Lemma 4.1.** *Under Assumption 1, the iterates of* Proj-PEG *with $\gamma > 0$ satisfy for any $k > 0$,*

$$\Psi_{k+1} \le \Psi_k - \left(1 - 5L^2\gamma^2\right)\|x^{k+1} - \widetilde{x}^k\|^2 - \gamma^2\|F(x^{k+1}) - F(\widetilde{x}^k)\|^2, \tag{29}$$

*where $\Psi_k = \|x^k - x^{k-1}\|^2 + \|x^k - x^{k-1} - 2\gamma(F(x^k) - F(\widetilde{x}^{k-1}))\|^2$.*

*Proof.* Since $F$ is monotone and $L$-Lipschitz, we have

$$0 \le \langle F(x^{k+1}) - F(x^k), x^{k+1} - x^k \rangle, \tag{30}$$
$$\|F(x^{k+1}) - F(\widetilde{x}^k)\|^2 \le L^2\|x^{k+1} - \widetilde{x}^k\|^2. \tag{31}$$

Taking into account relations $x^{k+1} = \text{proj}[x^k - \gamma F(\widetilde{x}^k)], x^k = \text{proj}[x^{k-1} - \gamma F(\widetilde{x}^{k-1})], \widetilde{x}^k = \text{proj}[x^k - \gamma F(\widetilde{x}^{k-1})]$ and properties of the projection on a convex closed set, we also obtain

$$\langle x^k - \gamma F(\widetilde{x}^k) - x^{k+1}, x^k - x^{k+1} \rangle \le 0, \tag{32}$$
$$\langle x^{k-1} - \gamma F(\widetilde{x}^{k-1}) - x^k, x^{k+1} - x^k \rangle \le 0, \tag{33}$$
$$\langle x^{k-1} - \gamma F(\widetilde{x}^{k-1}) - x^k, \widetilde{x}^k - x^k \rangle \le 0, \tag{34}$$
$$\langle x^k - \gamma F(\widetilde{x}^{k-1}) - \widetilde{x}^k, x^{k+1} - \widetilde{x}^k \rangle \le 0. \tag{35}$$

Summing up inequalities (30)-(35) with weights $4\gamma, 5\gamma^2, 4, 2, 2, 2$ respectively, we get

$$
\begin{aligned}
5\gamma^2\|F(x^{k+1}) - F(\widetilde{x}^k)\|^2 \le\ & 5\gamma^2 L^2\|x^{k+1} - \widetilde{x}^k\|^2 + 4\gamma\langle F(x^{k+1}) - F(x^k), x^{k+1} - x^k \rangle \\
& -4\langle x^k - \gamma F(\widetilde{x}^k) - x^{k+1}, x^k - x^{k+1} \rangle \\
& -2\langle x^{k-1} - \gamma F(\widetilde{x}^{k-1}) - x^k, x^{k+1} - x^k \rangle \\
& -2\langle x^{k-1} - \gamma F(\widetilde{x}^{k-1}) - x^k, \widetilde{x}^k - x^k \rangle \\
& -2\langle x^k - \gamma F(\widetilde{x}^{k-1}) - \widetilde{x}^k, x^{k+1} - \widetilde{x}^k \rangle \\
=\ & 5\gamma^2 L^2\|x^{k+1} - \widetilde{x}^k\|^2 + 4\gamma\langle F(x^{k+1}) - F(x^k), x^{k+1} - x^k \rangle \\
& -4\|x^{k+1} - x^k\|^2 + 4\gamma\langle F(\widetilde{x}^k), x^k - x^{k+1} \rangle \\
& +4\gamma\langle F(\widetilde{x}^{k-1}), x^{k+1} - x^k \rangle - 2\langle x^{k-1} - x^k, x^{k+1} - x^k \rangle \\
& -2\langle x^{k-1} - x^k, \widetilde{x}^k - x^k \rangle - 2\langle x^k - \widetilde{x}^k, x^{k+1} - \widetilde{x}^k \rangle \\
=\ & 5\gamma^2 L^2\|x^{k+1} - \widetilde{x}^k\|^2 + 4\gamma\langle F(x^{k+1}) - F(\widetilde{x}^k), x^{k+1} - x^k \rangle \\
& -4\|x^{k+1} - x^k\|^2 - 4\gamma\langle F(x^k) - F(\widetilde{x}^{k-1}), x^{k+1} - x^k \rangle \\
& +\|x^{k-1} - x^k\|^2 + \|x^{k+1} - x^k\|^2 - \|x^{k+1} + x^{k-1} - 2x^k\|^2 \\
& +\|x^{k-1} - \widetilde{x}^k\|^2 - \|x^{k-1} - x^k\|^2 - \|\widetilde{x}^k - x^k\|^2 \\
& +\|x^k - x^{k+1}\|^2 - \|x^k - \widetilde{x}^k\|^2 - \|x^{k+1} - \widetilde{x}^k\|^2.
\end{aligned}
$$

Next, we rearrange the terms and use $\Psi_k = \|x^k - x^{k-1}\|^2 + \|x^k - x^{k-1} - 2\gamma(F(x^k) - F(\widetilde{x}^{k-1}))\|^2 = 2\|x^k - x^{k-1}\|^2 - 4\gamma\langle x^k - x^{k-1}, F(x^k) - F(\widetilde{x}^{k-1}) \rangle + 4\gamma^2\|F(x^k) - F(\widetilde{x}^{k-1})\|^2$:

$$
\begin{aligned}
\Psi_{k+1} \le\ & -(1 - 5\gamma^2 L^2)\|x^{k+1} - \widetilde{x}^k\|^2 - 4\gamma\langle F(x^k) - F(\widetilde{x}^{k-1}), x^{k+1} - x^k \rangle \\
& -\|x^{k+1} + x^{k-1} - 2x^k\|^2 + \|x^{k-1} - \widetilde{x}^k\|^2 - 2\|\widetilde{x}^k - x^k\|^2 \\
& -\gamma^2\|F(x^{k+1}) - F(\widetilde{x}^k)\|^2 \\
=\ & \Psi_k - 2\|x^k - x^{k-1}\|^2 - 2\langle 2\gamma(F(x^k) - F(\widetilde{x}^{k-1})), x^{k+1} + x^{k-1} - 2x^k \rangle \\
& -4\gamma^2\|F(x^k) - F(\widetilde{x}^{k-1})\|^2 - (1 - 5\gamma^2 L^2)\|x^{k+1} - \widetilde{x}^k\|^2 \\
& -\|x^{k+1} + x^{k-1} - 2x^k\|^2 + \|x^{k-1} - \widetilde{x}^k\|^2 - 2\|\widetilde{x}^k - x^k\|^2 \\
& -\gamma^2\|F(x^{k+1}) - F(\widetilde{x}^k)\|^2.
\end{aligned}
$$

Using standard inequalities $2\langle a,b\rangle \leq \|a\|^2 + \|b\|^2$, $\|a+b\|^2 \leq 2\|a\|^2 + 2\|b\|^2$, $a,b \in \mathbb{R}^d$, we derive

$$
\begin{aligned}
\Psi_{k+1} &\leq \Psi_k - 2\|x^k - x^{k-1}\|^2 + 4\gamma^2\|F(x^k) - F(\widetilde{x}^{k-1})\|^2 + \|x^{k+1} + x^{k-1} - 2x^k\|^2 \\
&\quad -4\gamma^2\|F(x^k) - F(\widetilde{x}^{k-1})\|^2 - (1 - 5\gamma^2 L^2)\|x^{k+1} - \widetilde{x}^k\|^2 \\
&\quad -\|x^{k+1} + x^{k-1} - 2x^k\|^2 + 2\|x^{k-1} - x^k\|^2 + 2\|x^k - \widetilde{x}^k\|^2 - 2\|\widetilde{x}^k - x^k\|^2 \\
&\quad -\gamma^2\|F(x^{k+1}) - F(\widetilde{x}^k)\|^2 \\
&= \Psi_k - (1 - 5\gamma^2 L^2)\|x^{k+1} - \widetilde{x}^k\|^2 - \gamma^2\|F(x^{k+1}) - F(\widetilde{x}^k)\|^2,
\end{aligned}
$$

which finishes the proof. $\qquad\square$

We can now prove the main theorem.

**Theorem 2.** *Under Assumption 1, the iterates of* Proj-PEG *with $\gamma \leq 1/4L$ satisfy $\Phi_{k+1} \leq \Phi_k$, $k \geq 2$ with $\Phi_k$ defined as*

$$
\Phi_k = \|x^k - x^*\|^2 + \frac{1}{16}\|\widetilde{x}^{k-1} - \widetilde{x}^{k-2}\|^2 + \frac{3k + 32}{24}\Psi_k, \tag{36}
$$

*where $\Psi_k = \|x^k - x^{k-1}\|^2 + \|x^k - x^{k-1} - 2\gamma(F(x^k) - F(\widetilde{x}^{k-1}))\|^2$. In particular, it implies*

$$
\|x^N - x^{N-1}\|^2 \leq \frac{24 H_{0,\gamma}^2}{3N + 32}, \quad \mathit{Gap}_F(x^N) \leq \frac{8\sqrt{3} H_{0,\gamma} \cdot H_0}{\gamma\sqrt{3N + 32}}, \quad \forall N \geq 2, \tag{37}
$$

*where $H_0, H_{0,\gamma} > 0$ are such that $H_{0,\gamma}^2 = 2(1 + 3\gamma^2 L^2 + 4\gamma^4 L^4)\|x^0 - x^*\|^2 + \left(\frac{41}{12} + \frac{19}{3}\gamma^2 L^2\right)\gamma^2\|F(x^0)\|^2$, $H_0^2 = 3\|x^0 - x^*\|^2 + \frac{1}{30L^2}\|F(x^0)\|^2$.*

*Proof.* Lemma A.1 implies

$$
\begin{aligned}
0 &\leq 2\gamma\langle F(x^*), \widetilde{x}^k - x^*\rangle \leq 2\gamma\langle F(\widetilde{x}^k), \widetilde{x}^k - x^*\rangle \\
&\leq \|x^k - x^*\|^2 - \|x^{k+1} - x^*\|^2 - \|\widetilde{x}^k - x^k\|^2 + \gamma^2 L^2\|\widetilde{x}^k - \widetilde{x}^{k-1}\|^2.
\end{aligned}
$$

Together with Lemma A.2 it gives

$$
\begin{aligned}
\|x^{k+1} - x^*\|^2 + \frac{1}{16}\|\widetilde{x}^k - \widetilde{x}^{k-1}\|^2 &\leq \|x^k - x^*\|^2 - \|\widetilde{x}^k - x^k\|^2 + \gamma^2 L^2\|\widetilde{x}^k - \widetilde{x}^{k-1}\|^2 \\
&\quad + \frac{1}{16}\|\widetilde{x}^{k-1} - \widetilde{x}^{k-2}\|^2 - \frac{1}{16}\|\widetilde{x}^{k-1} - \widetilde{x}^{k-2}\|^2 \\
&\quad + \frac{1}{4}\|\widetilde{x}^k - x^k\|^2 + \frac{\gamma^2 L^2}{4}\|\widetilde{x}^{k-1} - \widetilde{x}^{k-2}\|^2 \\
&\quad - \frac{1}{16}\|\widetilde{x}^k - \widetilde{x}^{k-1}\|^2 \\
&= \|x^k - x^*\|^2 + \frac{1}{16}\|\widetilde{x}^{k-1} - \widetilde{x}^{k-2}\|^2 - \frac{3}{4}\|\widetilde{x}^k - x^k\| \\
&\quad - \frac{1 - 16\gamma^2 L^2}{16}\|\widetilde{x}^k - \widetilde{x}^{k-1}\|^2 \\
&\quad - \frac{1 - 4\gamma^2 L^2}{16}\|\widetilde{x}^{k-1} - \widetilde{x}^{k-2}\|^2 \\
&\leq \|x^k - x^*\|^2 + \frac{1}{16}\|\widetilde{x}^{k-1} - \widetilde{x}^{k-2}\|^2 - \frac{3}{4}\|\widetilde{x}^k - x^k\|^2,
\end{aligned}
$$

where in the last inequality we apply $\gamma \leq 1/4L$. Combining the above inequality with (29), we derive

$$
\begin{aligned}
\Phi_{k+1} \;\leq\; & \Phi_k + \frac{3}{24}\left(\|x^{k+1}-x^k\|^2 + \|x^{k+1}-x^k - 2\gamma(F(x^{k+1})-F(\widetilde{x}^k))\|^2\right) \\
& -\frac{3}{4}\|\widetilde{x}^k - x^k\|^2 - \frac{3k+32}{24}\left((1-5\gamma^2 L^2)\|x^{k+1}-\widetilde{x}^k\|^2 + \gamma^2\|F(x^{k+1})-F(\widetilde{x}^k)\|^2\right) \\
\;\leq\; & \Phi_k + \frac{1}{8}\left(3\|x^{k+1}-x^k\|^2 + 8\gamma^2\|F(x^{k+1})-F(\widetilde{x}^k)\|^2\right) \\
& -\frac{3}{4}\|\widetilde{x}^k - x^k\|^2 - \frac{4}{3}\left(\frac{9}{16}\|x^{k+1}-\widetilde{x}^k\|^2 + \gamma^2\|F(x^{k+1})-F(\widetilde{x}^k)\|^2\right) \\
\;\leq\; & \Phi_k + \underbrace{\left(\frac{6}{8}-\frac{4}{3}\cdot\frac{9}{16}\right)}_{0}\|x^{k+1}-\widetilde{x}^k\|^2 + \underbrace{\left(\frac{6}{8}-\frac{3}{4}\right)}_{0}\|\widetilde{x}^k - x^k\|^2 \\
& -\frac{\gamma^2}{3}\|F(x^{k+1})-F(\widetilde{x}^k)\|^2 \\
\;\leq\; & \Phi_k.
\end{aligned}
$$

In other words, we proved that $\Phi_k$ defined in (36) is a potential for (Proj-PEG). To prove the bounds from (37) using this potential, we need to derive several technical inequalities. Due to the non-expansiveness of the projection operator and Assumption 1, we have

$$
\begin{aligned}
\|x^1 - x^0\|^2 \;=\;& \|\operatorname{proj}[x^0 - \gamma F(x^0)] - \operatorname{proj}[x^0]\|^2 \\
\;\leq\;& \gamma^2\|F(x^0)\|^2, & (38) \\
\|x^1 - x^0 - 2\gamma(F(x^1)-F(\widetilde{x}^0))\|^2 \;=\;& \|x^1-x^0\|^2 - 4\gamma\langle x^1-x^0, F(x^1)-F(x^0)\rangle \\
& +4\gamma^2\|F(x^1)-F(x^0)\|^2 \\
\;\overset{(1)}{\leq}\;& \|x^1-x^0\|^2 + 4\gamma^2 L^2\|x^1-x^0\|^2 \\
\;\overset{(38)}{\leq}\;& (1+4\gamma^2 L^2)\gamma^2\|F(x^0)\|^2, & (39) \\
\Psi_2 \;\overset{(29)}{\leq}\; \Psi_1 \;\overset{(38),(39)}{\leq}\;& 2(1+2\gamma^2 L^2)\gamma^2\|F(x^0)\|^2. & (40)
\end{aligned}
$$

Next, using similar reasoning, we derive

$$
\begin{aligned}
\|\widetilde{x}^1 - \widetilde{x}^0\|^2 \;=\;& \|\operatorname{proj}[x^1-\gamma F(x^0)]-\operatorname{proj}[x^0]\|^2 \leq \|x^1-\gamma F(x^0)-x^0\|^2 \\
\;\overset{(21)}{\leq}\;& 2\|x^1-x^0\|^2 + 2\gamma^2\|F(x^0)\|^2 \;\overset{(38)}{\leq}\; 4\gamma^2\|F(x^0)\|^2, & (41) \\
\|x^1 - x^*\|^2 \;=\;& \|\operatorname{proj}[x^0-\gamma F(x^0)]-\operatorname{proj}[x^*-\gamma F(x^*)]\|^2 \\
\;\leq\;& \|x^0-\gamma F(x^0)-x^*+\gamma F(x^*)\|^2 \\
\;=\;& \|x^0-x^*\|^2 - 2\gamma\langle x^0-x^*, F(x^0)-F(x^*)\rangle + \gamma^2\|F(x^0)-F(x^*)\|^2 \\
\;\overset{(1)}{\leq}\;& (1+\gamma^2 L^2)\|x^0-x^*\|^2, & (42) \\
\|x^2 - x^*\|^2 \;=\;& \|\operatorname{proj}[x^1-\gamma F(\widetilde{x}^1)]-\operatorname{proj}[x^*-\gamma F(x^*)]\|^2 \\
\;\leq\;& \|x^1-\gamma F(\widetilde{x}^1)-x^*+\gamma F(x^*)\|^2 \;\overset{(21)}{\leq}\; 2\|x^1-x^*\|^2 + 2\gamma^2\|F(\widetilde{x}^1)-F(x^*)\|^2 \\
\;\overset{(1)}{\leq}\;& 2\|x^1-x^*\|^2 + 2\gamma^2 L^2\|\widetilde{x}^1-x^*\|^2 \\
\;=\;& 2\|x^1-x^*\|^2 + 2\gamma^2 L^2\|\operatorname{proj}[x^1-\gamma F(x^0)]-\operatorname{proj}[x^*-\gamma F(x^*)]\|^2 \\
\;\leq\;& 2\|x^1-x^*\|^2 + 2\gamma^2 L^2\|x^1-\gamma F(x^0)-x^*-\gamma F(x^*)\|^2 \\
\;\overset{(21)}{\leq}\;& 2\|x^1-x^*\|^2 + 4\gamma^2 L^2\|x^1-x^*\|^2 + 4\gamma^4 L^2\|F(x^0)-F(x^*)\|^2 \\
\;\overset{(42),(1)}{\leq}\;& 2(1+3\gamma^2 L^2 + 4\gamma^4 L^4)\|x^0-x^*\|^2. & (43)
\end{aligned}
$$

Therefore, for all $k \geq 2$ we have

$$
\begin{aligned}
\frac{3k+32}{24}\|x^k - x^{k-1}\|^2 \quad &\leq \quad \frac{3k+32}{24}\Psi_k \leq \Phi_k \leq \Phi_2 \\
&= \quad \|x^2 - x^*\|^2 + \frac{1}{16}\|\widetilde{x}^1 - \widetilde{x}^0\|^2 + \frac{19}{12}\Psi_2 \\
&\overset{(40),(41),(43)}{\leq} \quad 2(1 + 3\gamma^2 L^2 + 4\gamma^4 L^4)\|x^0 - x^*\|^2 \\
&\qquad\qquad + \left(\frac{41}{12} + \frac{19}{3}\gamma^2 L^2\right)\gamma^2\|F(x^0)\|^2 \\
&\overset{\text{def}}{=} \quad H_{0,\gamma}^2. \qquad\qquad\qquad\qquad\qquad\qquad (44)
\end{aligned}
$$

This implies the first part of (37). Moreover, using (44), we get $\|x^k - x^*\|^2 \leq \Phi_k \leq \Phi_2 \leq H_{0,\gamma}^2$ for all $k > 0$. Next, one can rewrite $\Psi_k$ as

$$
\begin{aligned}
\Psi_k &= \|x^k - x^{k-1}\|^2 + \|x^k - x^{k-1} - 2\gamma(F(x^k) - F(\widetilde{x}^{k-1}))\|^2 \\
&= 2\|x^k - x^{k-1} - \gamma(F(x^k) - F(\widetilde{x}^{k-1}))\|^2 + 2\gamma^2\|F(x^k) - F(\widetilde{x}^{k-1})\|^2
\end{aligned}
$$

that gives

$$
\|x^k - x^{k-1} - \gamma(F(x^k) - F(\widetilde{x}^{k-1}))\|^2 \leq \frac{\Psi_k}{2} \overset{(44)}{\leq} \frac{12 H_{0,\gamma}^2}{3k+32}. \qquad (45)
$$

Finally, for all $y \in \mathcal{X}$ we have

$$
0 \leq \langle x^{k-1} - \gamma F(\widetilde{x}^{k-1}) - x^k, x^k - y\rangle, \qquad (46)
$$

since $x^k = \text{proj}[x^{k-1} - \gamma F(\widetilde{x}^{k-1})]$. Putting all together, we derive

$$
\begin{aligned}
\text{Gap}_F(x^k) \quad &= \quad \max_{y \in \mathcal{X}: \|y-x^*\| \leq H_0} \langle F(y), x^k - y\rangle \\
&\overset{(1)}{\leq} \quad \max_{y \in \mathcal{X}: \|y-x^*\| \leq H_0} \langle F(x^k), x^k - y\rangle \\
&\overset{(46)}{\leq} \quad \frac{1}{\gamma} \max_{y \in \mathcal{X}: \|y-x^*\| \leq H_0} \langle x^{k-1} - x^k - \gamma(F(\widetilde{x}^{k-1}) - F(x^k)), x^k - y\rangle \\
&\leq \quad \frac{1}{\gamma}\|x^k - x^{k-1} - \gamma(F(x^k) - F(\widetilde{x}^{k-1}))\| \max_{y \in \mathcal{X}: \|y-x^*\| \leq H_0} \|x^k - y\| \\
&\overset{(45)}{\leq} \quad \frac{8\sqrt{3} H_{0,\gamma} \cdot H_0}{\gamma\sqrt{3k+32}}.
\end{aligned}
$$

This concludes the proof. $\qquad\qquad\qquad\qquad\qquad\qquad\qquad\qquad\qquad\qquad\qquad\qquad\qquad$ □

# D Further Numerical Experiments

In this section, we provide the base numerical results that grounded the results from Theorem 2. We also provide numerical experiments suggesting the same behavior for Optimistic Gradient (OG) in the constrained case (as provided by, e.g., [Hsieh et al., 2019, Section 3]) whose recursion is:

$$\widetilde{x}^k = \text{proj}[x^k - \gamma F(\widetilde{x}^{k-1})], \quad x^{k+1} = \widetilde{x}^k + \gamma(F(\widetilde{x}^{k-1}) - F(\widetilde{x}^k)), \quad \text{for all } k > 0, \quad \text{(Proj-OG)}$$

with $\widetilde{x}^0 = x^0 \in \mathcal{X}$. We report the worst-case evolution of the ratios $\|x^N - x^{N-1}\|^2 / \|x^0 - x^*\|^2$ (for Proj-PEG) and $\|\widetilde{x}^N - \widetilde{x}^{N-1}\|^2 / \|x^0 - x^*\|^2$ (for Proj-OG) on Fig. 2. Those values were computed using the performance estimation toolbox (PESTO) [Taylor et al., 2017b].

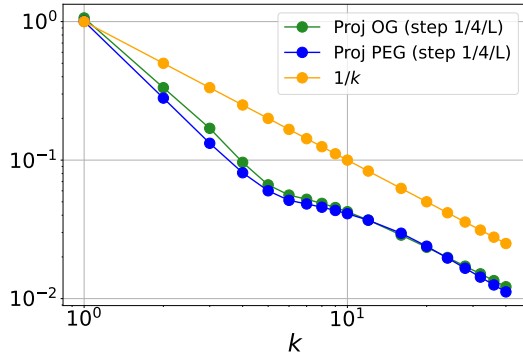

Figure 2: Worst-case ratios $\|x^N - x^{N-1}\|^2 / \|x^0 - x^*\|^2$ (for Proj-PEG) and $\|\widetilde{x}^N - \widetilde{x}^{N-1}\|^2 / \|x^0 - x^*\|^2$ (for Proj-OG) as functions of $N$, computed with PESTO [Taylor et al., 2017b] ($L = 1$, $\gamma = 1/4L$).