# OpenReview forum: "Last-Iterate Convergence of Optimistic Gradient Method for Monotone Variational Inequalities"
_NeurIPS.cc/2022/Conference — NeurIPS 2022 Accept_

### Official Review · Reviewer_r8QB · 2022-07-09

**Rating:** 5
**Confidence:** 4
**Soundness:** 3 good
**Presentation:** 4 excellent
**Contribution:** 2 fair

**Summary:**

Optimistic Gradient Method (OGM) is a variant of Extra Gradient (EG) that requires one gradient per iteration instead of two. OGM can be used to in min-max problems such as game optimization, adversarial learning, and GAN training. This paper considers VIP with smooth and monotone operator and proves that for constrained and unconstrained cases the last iterate of OGM converges to the solution set sublinearly. The paper leverages ​​Performance Estimation Problems (PEP) to get an idea to build the proper Lyupnov functions and then prove the convergence of these functions.

**Questions:**

1- It seems that this paper already sovled this problem with similar assumptions:
“Tight Last-Iterate Convergence of the Extragradient and the Optimistic Gradient Descent-Ascent Algorithm for Constrained Monotone Variational Inequalities”. It would be helpful if you explain the difference of your result and this paper.

2- In line 96-96 you claim that PEG is still largely used in practice. I suggest to add some citations to a real-world models that use PEG.

3- In Line 238 it should be non-positive.


**Limitations:**

This works addresses the potential social impacts.


**Strengths And Weaknesses:**

Clarity

The paper properly explains the difference between OGM and EG and the analysis technique their use e.g. PEP and when compares two variants of OGM.

Originality

Last iterate convergence proof is not new and I think this paper contribution is a incremental one. However their theoretical comparison between variants of OGM was interesting.


Quality

All the proofs presented were correct and sound. The paper is a theoretical mainly however their experimental result in appendix D is not explained well.


Significance

Last iterate convergence is important since in practice the last iterate is usually used as the final result not the averaged one.

Limitation

The major limitation of this work is that it doesn’t consider the stochastic setting that is mostly used in practice for training large machine learning models with enormous data.

---

> ### Author Response · Authors · 2022-08-01
> **Authors' response to Official Review of Paper9014 by Reviewer r8QB [Part 2/2]**
>
> > In line 96-96 you claim that PEG is still largely used in practice. I suggest to add some citations to a real-world models that use PEG.
>
> Thank you for your suggestion. For example, Daskalakis et al., [2018] show that PEG-based algorithms (like PEG-Adam) perform well in training WGAN on CIFAR10. Next, PEG and OG have been extensively used in regret matching [Brown & Sandholm, 2019], counterfactual regret minimization [Farina et al., 2019] with applications such as training agents to play poker [Anagnostides et al., 2022]. We will add these remarks along with the missing references in the final version of our work.
>
> ---
>
> ### References:
>
> Brown, N., & Sandholm, T. (2019, July). Solving imperfect-information games via discounted regret minimization. In Proceedings of the AAAI Conference on Artificial Intelligence (Vol. 33, No. 01, pp. 1829-1836).
>
> Farina, G., Kroer, C., Brown, N., & Sandholm, T. (2019, May). Stable-predictive optimistic counterfactual regret minimization. In International conference on machine learning (pp. 1853-1862). PMLR.
>
> Anagnostides, I., Panageas, I., Farina, G., & Sandholm, T. (2022). On Last-Iterate Convergence Beyond Zero-Sum Games. arXiv preprint arXiv:2203.12056.
>
> ---
>
> >  In Line 238 it should be non-positive.
>
> Thank you very much, we will fix it.

---

> ### Author Response · Authors · 2022-08-01
> **Authors' response to Official Review of Paper9014 by Reviewer r8QB [Part 1/2]**
>
> We want to thank the Reviewer for their time on the paper. We are pleased to read that the reviewer even took the time to read our appendices. Below are our answers to the Reviewer's questions. We hope we have addressed and resolved all of their concerns.
>
> ---
>
> > Last iterate convergence proof is not new and I think this paper contribution is a incremental one.
>
> **A:** It is unclear to us what the reviewer means by “incremental”. Although we clearly build on the literature on this topic, the results contained in this paper are new, to the best of our knowledge. That is, unlike previous work [Golowich et al., 2020a], we: (i) do not assume Lipschitzness of the Jacobian, and (ii) consider constrained setup. Furthermore, our analyses significantly differ from the one by [Golowich et al., 2020a] (see, e.g., lines 259 - 264 for a summary).
> As explained in Section 2, obtaining last-iterate convergence results for PEG appears to be a nontrivial task because of at least two aspects: (1) $||F(x^k)||$ does not decrease monotonically, and (2) it is crucial to use PEG form instead of OG form to get a “simple” proof.
>
> ---
>
> > It seems that this paper already solved this problem with similar assumptions: “Tight Last-Iterate Convergence of the Extragradient and the Optimistic Gradient Descent-Ascent Algorithm for Constrained Monotone Variational Inequalities”. It would be helpful if you explain the difference of your result and this paper.
>
> **A:** Thank you for your comment. During the preparation of this work, we were not aware of the mentioned paper, a concurrent work that was carried out independently. However, we do mention its first version in our text. As mentioned in, e.g., https://nips.cc/Conferences/2020/PaperInformation/ReviewerGuidelines (Point 6), we believe that our submission should be considered contemporaneous to this preprint. Furthermore, the first simplified version of this concurrent work appeared in a later revision. More precisely, the first draft of the concurrent work appeared on April 20th and did not contain either simple proofs or any result on OG. The versions that are mentioned by Reviewer NX5e appear on May 10th and 16th, i.e., less than a week before the NeurIPS submission deadline. Thus, it is clear that our work has been developed strictly independently from Cai et al [2022].
>
> We now provide a detailed comparison that we will incorporate in the final version of our work:
>
> Initially, [Cai et al, 2022] considered SOS programs with polynomials of degrees higher than 2, thereby making their proofs artificially complicated (compared to degree-2 certificate). Their paper also contained a few erroneous statements concerning the impossibility of obtaining degree-2 certificates (i.e., applying PEPs). The authors simplified their proofs after a revision and obtained similar results to the ones presented in this submission using SOS with degree-2 polynomials, i.e., PEPs.
> That being said, their results rely on the tangent residual, and they analyze OG in the constrained case. As a result, they have a different potential/Lyapunov function and arguably more complicated proofs. For example, in Lemma 4.1, we use *six* inequalities, while in the mentioned paper, the authors combine *nine* inequalities to prove the monotonicity of their potential/Lyapunov function.
>
> ---
>
>
> > The paper is a theoretical mainly however their experimental result in appendix D is not explained well.
>
> **A:** The main goal of this experiment is to show that Proj-OG also has $\mathcal{O}(1/K)$ last-iterate convergence rate, and this rate can be obtained via PEP, thereby relying on the same technique as in the core paper, so there is no other ingredient to be understood. We will make this more transparent in the final version of the paper. For clarity, we will provide all codes for reproducing our results, as well.
>
> ---
>
> > The major limitation of this work is that it doesn’t consider the stochastic setting that is mostly used in practice for training large machine learning models with enormous data.
>
> **A:** Thank you for your comment again. From what we can tell, establishing last-iterate convergence was already a nontrivial task in the deterministic setting. Although EG dates back to 1976, the results on its last-iterate convergence were only obtained in 2021. Similarly, OG was developed in 1980, and its last-iterate convergence was still open. Although we acknowledge the importance of the stochastic setting, we do believe it is reasonable to understand the deterministic setting before the stochastic one.

---

> ### Author Response · Authors · 2022-08-09
> **Request to Reviewer r8QB**
>
> Dear Reviewer r8QB,
>
> We thank you once again for your feedback and time. Since the discussion period ends today, we kindly remind that we will be happy to hear what do you think about our response to your questions and concerns.
>
> Kind regards,
>
> Authors of Paper 9014

---

### Official Review · Reviewer_DFRa · 2022-07-12

**Rating:** 6
**Confidence:** 3
**Soundness:** 4 excellent
**Presentation:** 4 excellent
**Contribution:** 3 good

**Summary:**

The paper studies the Past Extragradient (PEG) algorithm for solving variational inequalities wrt to a monotone Lipschitz vector functional F. The main theorem of the paper derivesa $O(1/N)$ convergence rates for $||F(x_n)||^2$ in unconstrained problems and same rate for $||x_n-x_{n-1}||^2$ for constrained problems; and further show $O(1/\sqrt(N))$ convergence in the Gap functions.

The overview of the proof is given by numerically bounding the worst-case convergence in terms of a sdp relaxation or appropriately parametrized performance estimation problems. The main lemma of the proof establishes a descent criteria in terms of a chosen potential function, which in turn establishes the convergence rates. The closest contribution (based on related work stated in the paper) shows a similar rate for PEG with different proof technique and stronger assumption on $\nabla F$ being Lipschitz.

I am not fully familiar with related work in this space, but the main results and proofs of the papers are stated clearly.  To my knowledge the proofs in the appendix appear right and the theorem is interesting for the neurips community.



**Questions:**

See comment above on comparison to EG.

**Strengths And Weaknesses:**

The paper is well written (I didn't even find many minor typos) and was accessible to me even though I was not familiar with the literature.

One comment/question I do have is the following: In the related work, the authors claim that Gorbunov et al. 2021 and Cai et al. 2022 show convergence rate of O(1/N) for the related EG algorithm under the same settings of the current paper. From my brief exploration, it appears PEG is a slightly different algorithm, and the analysis is valuable as is. However, it would be useful to reach the broader audience if the authors could comment on what kind of problem/compute instances would benefit from the use of PEG over EG and how different are these algorithms and the proof technique from Gorbunov et al. 2021? The authors already provide such discussion when compared to other related work like Golowich 2020a and EG with anchoring.

Typo: Line 188, there should be no tilde on top of G_PEG and G_OG.

Additional comment, as a closely related work it would be helpful to mention the updates for EG somewhere in the paper. For example, in lines 27-28, both algorithms are mentioned but only one is defined which makes a parsing of the sentence a bit hard.

---

> ### Author Response · Authors · 2022-08-01
> **Authors' response to Official Review of Paper9014 by Reviewer DFRa**
>
> We would like to thank the Reviewer for the time spent closely reading our submission. Below is our answer to their comment. We hope we have addressed and resolved all of their concerns.
>
> ---
>
> >  In the related work, the authors claim that Gorbunov et al. 2021 and Cai et al. 2022 show convergence rate of O(1/N) for the related EG algorithm under the same settings of the current paper. From my brief exploration, it appears PEG is a slightly different algorithm, and the analysis is valuable as is. However, it would be useful to reach the broader audience if the authors could comment on what kind of problem/compute instances would benefit from the use of PEG over EG, how different are these algorithms and the proof technique from Gorbunov et al. 2021?
>
> **A:** We thank the reviewer for this great comment. PEG and EG are two slightly different algorithms for Variational inequality Problems. Even if these two algorithms are very similar from an implementation perspective, their last-iterate convergence analysis is drastically different. As an illustration, Gorbunov et al. [2021] shows that the quantity $||F(x^k)||$ is decreasing for EG (which has a nice interpretation), while as far as PEG is concerned, neither $||F(x^k)||$ nor $||x^k - x^{k-1}||$ are monotonically decreasing, rendering the analysis more complicated. This is the reason why we had to find different potentials such as those given in Lemmas 3.1 and 4.1. Moreover, Gorbunov et al. [2021] do not study the constrained case, which turned out to be a nontrivial extension of the unconstrained one.
>
> Moreover, we mention the benefits of using PEG in lines 98-103. In particular, unlike EG, PEG can be applied to online learning and only require one gradient evaluation per iteration.
>
> ---
>
> > Typo: Line 188, there should be no tilde on top of G_PEG and G_OG.
>
> **A:** Thank you for your remark again. It turns out that this is not a typo. We put a tilde on top to emphasize that this formulation is a relaxation of the previous problem. That is, instead of requiring the existence of a monotone and Lipschitz operator $F$, we use (necessary but not sufficient) inequalities (7)-(8), thereby producing a relaxation (i.e., upper bounds) of the original problem.
>
> ---
>
> > Additional comment, as a closely related work it would be helpful to mention the updates for EG somewhere in the paper. For example, in lines 27-28, both algorithms are mentioned but only one is defined which makes a parsing of the sentence a bit hard.
>
> **A:** Thank you for the suggestion. We will add the update rule of EG in the final version of our work.

---

> > ### Comment · Reviewer_DFRa · 2022-08-06
> > **Comparison to EG vs EG with anchoring**
> >
> > Thanks for the response.
> >
> > To my understanding, lines 98-103 show comparison of PEG to EG+anchoring (as was also mentioned in my review) but in my review, the query was about comparison of PEG and EG itself -- not sure if this evident for experts, but as a reader unfamiliar with exact equations but familiar with the math and core problem, it was not fully evident from the paper when PEG would be preferable. This could also be helped with adding updates of EG and EG+anchoring and discussing the relation a bit more to make the paper self contained.

---

> > > ### Author Response · Authors · 2022-08-06
> > > **Thank you for your comment!**
> > >
> > > We thank the reviewer for the great comment and suggestions. We will add the exact update rules of EG and EG with anchoring to the final version of our work to make it self contained, as reviewer suggested. We will also add the following comparison of PEG and EG.
> > >
> > > The difference from the theoretical perspective between EG and PEG is described in our response and we will include this clarification in the final version. The main algorithmic difference between PEG and EG is that PEG requires only one oracle call per iteration while EG needs two oracle calls, which means that the the method is about two times longer to run. In practice, this difference might be significant. Moreover, since EG requires two oracle calls per iteration it cannot be applied to online learning in its pure form [Golowich et al., 2020]. The same issue is with Anchored EG.
> > >
> > > Moreover, we would like to mention that Daskalakis et al., [2018] show that PEG-based algorithms (like PEG-Adam) perform well in training WGAN on CIFAR10. Next, PEG and OG have been extensively used in regret matching [Brown & Sandholm, 2019], counterfactual regret minimization [Farina et al., 2019], and for training agents to play poker [Anagnostides et al., 2022]. We will add these remarks along with the missing references in the final version of our work.
> > >
> > > ---
> > >
> > > References:
> > >
> > > Brown, N., & Sandholm, T. (2019, July). Solving imperfect-information games via discounted regret minimization. In Proceedings of the AAAI Conference on Artificial Intelligence (Vol. 33, No. 01, pp. 1829-1836).
> > >
> > >
> > > Golowich, N., Pattathil, S., & Daskalakis, C. (2020). Tight last-iterate convergence rates for no-regret learning in multi-player games. Advances in neural information processing systems, 33, 20766-20778.
> > >
> > > Farina, G., Kroer, C., Brown, N., & Sandholm, T. (2019, May). Stable-predictive optimistic counterfactual regret minimization. In International conference on machine learning (pp. 1853-1862). PMLR.
> > >
> > > Anagnostides, I., Panageas, I., Farina, G., & Sandholm, T. (2022). On Last-Iterate Convergence Beyond Zero-Sum Games. arXiv preprint arXiv:2203.12056.

---

> > > > ### Comment · Reviewer_DFRa · 2022-08-08
> > > > **Thanks for the clarification**
> > > >
> > > > I don't have further comments. Based on discussion and other reviews, I would like to keep my score and recommend acceptance.

---

> > > > > ### Author Response · Authors · 2022-08-08
> > > > > **Thank you!**
> > > > >
> > > > > We thank the reviewer for their time, valuable feedback, and positive evaluation of our work.

---

### Official Review · Reviewer_NX5e · 2022-07-13

**Rating:** 6
**Confidence:** 4
**Soundness:** 4 excellent
**Presentation:** 4 excellent
**Contribution:** 3 good

**Summary:**

This paper analyzes the last-iterate convergence rates of the past extragradient (PEG) method, also known as the optimistic gradient (OG) method, for monotone variational inequality with/without a constraint set, which were not known previously. The authors leverage the performance estimation problem (PEP) technique that outputs numerical convergence rate bounds given problem parameters, step sizes, and the number of iterations. This, however, does not provide an analytical bound, and directly finding analytical bounds via PEP was impossible for the PEG. Instead, the authors cleverly first identified useful inequalities in Lemmas 3.1 and 4.1 for the unconstrained and constrained cases, respectively. These show that some quantities, other than the gradient norm, are monotonically nonincreasing,  (Note that, for the standard EG, the gradient norm was found to decrease monotonically, and this was the key for proving its $O(1/N)$ last-iterate convergence rate in [Gorbunov et al., 2021].) Lemmas 3.1 and 4.1 are then used for constructing potential functions that yielded analytical $O(1/N)$ bounds for the (projected) PEG.

**Questions:**

- The abstract seems to imply that the O(1/N) rate of the EG-type method without the Lipschitz Jacobian condition is not known before this paper. This is later clarified but I think the abstract can be improved so that there is no confusion.
- Are the derived bounds close to the exact rate bounds (found by PEP)?



**Limitations:**

Yes, the authors mentioned extensions to stochastic and non-monotone cases.

**Strengths And Weaknesses:**

**Strengths**
- Last iterate convergence: Last iterate is the one chosen in practice, so understanding its behavior is of interest, especially for the widely used OG method.
- Example 1.1: an interesting example that supports this paper's further study on PEG over recent developments on accelerated EG methods.
- Clever use of PEP: By using the PEP, the authors numerically identified the trends that (1) PEG has O(1/N) rate, (2) PEG form, rather than OG form, is necessary for the "simple" proof, and (3) the gradient norm does not decrease monotonically. Then, the authors used PEP to identify some quantities that monotonically decrease, which is the key of this paper's $O(1/N)$ rate proof.
- Fixed point residual rate: The rate results of the Proj-PEG are given in terms of standard optimality measure (the fixed point residual), whereas the only other work [Cai et al., 2022] found a rate of EG in terms of the (non-standard) tangent residual.
**Weaknesses**
- Somewhat similar results on EG are recently found in [Cai et al., 2022]
- Lack of intuition and explanation on two different potential functions, especially the one for the constrained case.

---

> ### Author Response · Authors · 2022-08-01
> **Authors' response to the Official Review of Paper9014 by Reviewer NX5e**
>
> We thank the reviewer for their careful assessment of our work and their encouraging words! We provide a few clarifications below.
>
> ---
> > Somewhat similar results on EG are recently found in [Cai et al., 2022]
>
> **A:** Thank you for your comment. During the preparation of this work, we were not aware of the mentioned paper, a concurrent work that was carried out independently. However, we do mention its first version in our text. As mentioned in, e.g., https://nips.cc/Conferences/2020/PaperInformation/ReviewerGuidelines (Point 6), our submission should be considered contemporaneous to this preprint. Furthermore, the first simplified version of this concurrent work appeared in a later revision. More precisely, the first draft of the concurrent work appeared on April 20th and contained neither simple proofs nor any result on OG. The versions that are mentioned by Reviewer NX5e appear on May 10th and 16th, i.e., less than one week before the NeurIPS submission deadline. Thus, it is clear that our work has been developed strictly independently from Cai et al [2022].
>
> We now provide a detailed comparison that we will incorporate in the final version of our work:
>
> Initially, [Cai et al, 2022] considered SOS programs with polynomials of degrees higher than 2, thereby making their proofs artificially complicated (compared to degree-2 certificates). Their paper also contained a few erroneous statements concerning the impossibility of obtaining degree-2 certificates (i.e., applying PEPs). The authors simplified their proofs after a revision and obtained similar results to the ones presented in this submission using SOS with degree-2 polynomials, i.e., PEPs.
>
> That being said, their results rely on the tangent residual, and they analyze OG in the constrained case. As a result, they have a different potential/Lyapunov function and arguably more complicated proofs. For example, in Lemma 4.1, we use *six* inequalities, while in the mentioned paper, the authors combine *nine* inequalities to prove the monotonicity of their potential/Lyapunov function.
>
> ---
>
> > Lack of intuition and explanation on two different potential functions, especially the one for the constrained case.
>
> **A:** Thank you again for your constructive comment! Although we agree with the reviewer that we do not provide any geometric intuition about our potential, we do disagree with the fact we do not provide any intuition.
>
> First, this lack of geometric intuition applies to most results relying on Lyapunov/potential arguments.
>
> Second, the fact that this convergence result was not known was probably at least partially due to the lack of intuitions for constructing such Lyapunov/potential analyses for this setting. Finally, our use of semidefinite programming is, in our sense, relatively straightforward. So whereas there is no clear geometric intuition, there is an algebraic intuition/construction: we explicitly look for proofs that can be formulated as linear combinations of inequalities.
>
> In other words, we searched for the Lyapunov/potential in Lemmas 3.1 and 4.1 via solving specifically constructed SDPs. Since the goal was to derive the rates for $||F(x^k)||^2$ and $||x^k - x^{k-1}||^2$, we searched for potentials containing these terms. However, since $||F(x^k)||^2$ and $||x^k - x^{k-1}||^2$ are not necessarily non-increasing quantities, the potential has to contain extra terms as well. In the end, we present the simplest potentials we found. We point out that the second term in the potential from Lemma 4.1 is also very useful to derive the upper bound for $\text{Gap}_{F}(x^N)$ (see lines 505 - 507).
>
> ---
>
> > The abstract seems to imply that the O(1/N) rate of the EG-type method without the Lipschitz Jacobian condition is not known before this paper. This is later clarified but I think the abstract can be improved so that there is no confusion.
>
> **A:** We are not sure what the reviewer precisely means here. From what we can tell, those facts were indeed not known for PEG before this work (and concurrent work from [Cai et al. 2022]).
>
> ---
>
> > Are the derived bounds close to the exact rate bounds (found by PEP)?
>
>
> **A:** Thank you for this sharp question. To be very accurate, PEPs do not provide an exact rate bound for this setup: they also provide upper bounds (as mentioned in lines 145-146, in this setting, the SDP formulations are relaxations of the problem of computing the worst-case). Furthermore, our goal was to identify and prove the rate, but we never really paid much attention to the constants. That being said, we will add plots illustrating the differences between the numerical bounds from PEP and our findings.

---

> > ### Comment · Reviewer_NX5e · 2022-08-08
> > **Thanks for the detailed response.**
> >
> > - While reading the authors' response to reviewer DRFa, it reminded me papers that study the anchored versions of PEG, which might be worth citing in the paper.
> >
> > Quoc Tran-Dinh and Yang Luo, Halpern-Type Accelerated and Splitting Algorithms For Monotone Inclusions, arXiv, 2021
> >
> > Quoc Tran-Dinh, The Connection Between Nesterov’s Accelerated Methods and Halpern Fixed-Point Iterations, arXiv, 2022
> >
> > In addition, I want to mention that, although PEG is preferable in terms of required oracles per iterations, one should not neglect the fact that it requires a smaller step size for convergence in theory and has a larger constant for the worst case rates.
> >
> > - Thanks for clarifying the timeline of this paper and [Cai et al., 2022]. I agree with the authors that [Cai et al., 2022] should be considered a concurrent work. Please ignore my comment on the abstract, which considers [Cai et al., 2022] as a preceding work.
> >
> > I have no further concerns on this paper. I think this is a good paper, and I maintain my score.

---

> > > ### Author Response · Authors · 2022-08-08
> > > **Thank you for your comment!**
> > >
> > > We thank the reviewer for pointing out very relevant papers and the remark about smaller stepsizes in theory and larger constant in the rate. We will cite these works in the final version of our paper. We will also add the discussion about differences between known rates for PEG and EG in terms the maximal possible theoretical stepsize and numerical constants in the convergence rates.

---

### Author Response · Authors · 2022-08-01
**General response to the reviewers**

We thank the reviewers for the time spent reading our work and for the constructive feedback. From what we can tell, the general evaluation is mostly positive. In summary, the reviewers identified the following strengths of our work:
- The last iterate is the one chosen in practice, so understanding its behavior is of interest and importance.
- Example 1.1 supports this paper's further study on PEG over recent developments in accelerated EG methods.
- By using the PEP, the authors numerically identified the trends that (1) PEG has an O(1/N) rate, (2) PEG form, rather than OG form, is - necessary for the "simple" proof, and (3) the gradient norm does not decrease monotonically.
- PEPs allow comparing variants of the same algorithm.
- Fixed point residual rate: The rate results of the Proj-PEG are given in terms of standard optimality measure (the fixed point residual), whereas the only other work [Cai et al., 2022] found a rate of EG in terms of the (non-standard) tangent residual.

Finally, the reviewers seemed to appreciate the writing style of the paper as well as its presentation and soundness.

That being said, there seems to be some concern about the closeness between our results and those of [Cai et al., 2022]. In short, as mentioned in the NeurIPS 2020 [reviewer guidelines (Point 6)](https://nips.cc/Conferences/2020/PaperInformation/ReviewerGuidelines) "Papers (whether refereed or not) appearing less than two months before the submission deadline are considered contemporaneous to NeurIPS submissions."

Thus, we believe that our submission should be considered contemporaneous to this preprint. Essentially, the first version of the preprint dates back to one month before the NeurIPS deadline and did not contain any result on optimistic gradient: it studies the extragradient method. The authors of [Cai et al, 2022] added results similar to those presented in this submission only a few days before the NeurIPS deadline (between May 10th and May 16th). We claim that our results are totally independent of theirs and that we had most of them a few months before the deadline. Finally, let us mention that the version [Cai et al, 2022] from April 20th contained a few erroneous statements and perhaps somewhat artificially complicated proofs. We discuss this in more detail in the answers to the individual reviewers.

In the responses below, we address all the concerns and questions raised by the reviewers.

---

### Meta-Review · Area_Chair_rP8k · 2022-08-25

**Recommendation:** Accept
**Confidence:** Less certain

**Metareview:**

Overall all reviewers were positive about this paper and I tend to agree, but no reviewer felt particularly excited about the results.

**Award:**

No

---

### Decision · Program_Chairs · 2022-09-14

Accept